# Enhanced carbonyl stress induces irreversible multimerization of CRMP2 in schizophrenia pathogenesis

Manabu Toyoshima[1,*] , Xuguang Jiang[2,*] , Tadayuki Ogawa[2] , Tetsuo Ohnishi[1] , Shogo Yoshihara[2] , Shabeesh Balan[1] , Takeo Yoshikawa[1] , Nobutaka Hirokawa[2,3]

**Enhanced carbonyl stress underlies a subset of schizophrenia, but its causal effects remain elusive. Here, we elucidated the molecular mechanism underlying the effects of carbonyl stress in iPS cells in which the gene encoding zinc metalloenzyme glyoxalase I (*GLO1*), a crucial enzyme for the clearance of carbonyl stress, was disrupted. The iPS cells exhibited significant cellular and developmental deficits, and hyper-carbonylation of collapsing response mediator protein 2 (CRMP2). Structural and biochemical analyses revealed an array of multiple carbonylation sites in the functional motifs of CRMP2, particularly D-hook (for dimerization) and T-site (for tetramerization), which are critical for the activity of the CRMP2 tetramer. Interestingly, carbonylated CRMP2 was stacked in the multimer conformation by irreversible cross-linking, resulting in loss of its unique function to bundle microtubules. Thus, the present study revealed that the enhanced carbonyl stress stemmed from the genetic aberrations results in neurodevelopmental deficits through the formation of irreversible dysfunctional multimer of carbonylated CRMP2.**

## Introduction

Schizophrenia is one of the most serious psychiatric disorders, with a prevalence of ~1% worldwide. Impairments in neurodevelopmental process are thought to play a pivotal role in the pathogenesis of schizophrenia ("neurodevelopmental theory of schizophrenia") (Stefansson et al, 2009; Insel, 2010; van Os et al, 2010). To date, oxidative stress and redox dysregulation have been identified as cogent pathophysiological factors for schizophrenia (Schulz et al, 2000; Young et al, 2007) and, thus, investigating the molecular mechanism underlying these events should identify advanced strategies for the development of innovative treatments and drugs.

Oxidative stress, which has been suggested to be crucial for the pathogenesis of schizophrenia, mainly causes oxidative damage to the lipids, proteins, and DNA (Bitanihirwe & Woo, 2011). It also forms one of the central hub systems, which when perturbed affects the integrity of parvalbumin interneurons and oligodendrocytes, a characteristic feature observed in schizophrenia (Steullet et al, 2016). Oxidative stress leverages the formation of advanced glycation end (AGE) products, which is also called "carbonyl stress." In cells, reactive carbonyl compounds (RCCs) are formed from sugars, lipids, and amino acids. The $\alpha$-oxoaldehydes methylglyoxal, glyoxal, and 3-deoxyglucosone are typical toxic RCCs, and their accumulation results in the production of AGEs, such as pentosidine (PEN). In contrast, pyridoxamine, an active form of vitamin B6 against carbonyl stress, is known to detoxify RCCs via carbonyl-amine chemistry (Itokawa et al, 2014). The activity of zinc metalloenzyme glyoxalase I (GLO1) plays an enzymatically central role in the removal of AGEs. Recent studies have reported that elevated carbonyl stress is relevant for the pathophysiologies of not only systemic diseases such as diabetes mellitus (Perkins et al, 2012) but also neuropsychiatric disorders, including schizophrenia, mood disorder, autism, and Alzheimer's disease (Junaid et al, 2004; LoPachin et al, 2007; Itokawa et al, 2014). In fact, 20% of schizophrenia patients reportedly exhibit increased blood levels of PEN and lowered levels of vitamin B6 (Arai et al, 2010; Miyashita et al, 2013) and share clinical features of treatment-resistance, a higher proportion of inpatients, a low educational status, longer durations of hospitalization, and higher doses of antipsychotic medications (Miyashita et al, 2013, 2014). Furthermore, the administration of pyridoxamine to patients with schizophrenia under enhanced carbonyl stress can effectively ameliorate their symptoms (Arai et al, 2014; Itokawa et al, 2018). Moreover, the offspring from diabetic mothers, who should be afflicted with elevated carbonyl stress, showed a sevenfold increased risk for schizophrenia (Van Lieshout & Voruganti, 2008), which suggested that carbonyl stress during brain development could predispose an individual to disease manifestation with a substantial effect size. Importantly, elevated carbonyl stress was observed even in postmortem brains from a subset of schizophrenia patients (Ohnishi et al, 2019).

In cells, microtubules (MTs) are highly dynamic 25-nm tubule-like polymers composed of $\alpha$, $\beta$-tubulin heterodimers and serve as

[1]Laboratory for Molecular Psychiatry, RIKEN Center for Brain Science, Wako, Japan   [2]Department of Cell Biology and Anatomy, Graduate School of Medicine, University of Tokyo, Tokyo, Japan   [3]Center of Excellence in Genome Medicine Research, King Abdulaziz University, Jeddah, Saudi Arabia

Correspondence: takeo.yoshikawa@riken.jp; hirokawa@m.u-tokyo.ac.jp
*Manabu Toyoshima and Xuguang Jiang contributed equally to this work

the cytoskeletal structures for cell morphogenesis and the rails for intracellular transport by motor proteins, including kinesin superfamily motor proteins (KIFs) (Hirokawa, 1998; Hirokawa et al, 2009). Therefore, MTs are fundamental structures for cells, and their dynamics are strictly controlled via MT-associated proteins and MT-depolymerizing proteins. Collapsing response mediator protein 2 (CRMP2 and DPYSL2) is known to be a member of the CRMP family. CRMP2 reportedly functions to regulate synaptic transmission, control MT dynamics, and modulate vesicle trafficking in cells and the developing brain (Inagaki et al, 2001; Charrier et al, 2003; Cole et al, 2004; Quach et al, 2004). In particular, CRMP2 has been implicated in crucial functions for neurite outgrowth, neuronal polarity with axon/dendrite specification and KIF-dependent vesicle transport (Arimura et al, 2000; Cole et al, 2004; Quach et al, 2004; Kawano et al, 2005; Brittain et al, 2009; Hensley et al, 2011). To date, the mechanisms underlying the function of CRMP2, particularly its activity in the regulation of MT polymerization and stabilization have been widely investigated. The CRMP2–tubulin interaction reportedly promotes MT polymerization through its GTPase-activating protein (GAP) activity and results in neurite outgrowth (Fukata et al, 2002; Chae et al, 2009). Post-translational modifications (PTMs), including phosphorylation and glycosylation, have also been reported to regulate CRMP2 activity (Zhang & Koch, 2017). CRMP2 was recently reported to promote MT bundling through its carboxy terminus, and its phosphorylation decreases its bundling activity (Zheng et al, 2018).

CRMP2, encoded by the *DPYSL2* gene, maps to the chromosome 8p21.2, which has been identified as schizophrenia susceptibility loci in genome-wide linkage studies (Ng et al, 2008). Subsequently, genetic association studies in different ethnicities have consistently shown the role of *DPYSL2* in schizophrenia susceptibility (Nakata et al, 2003; Fallin et al, 2005, 2011; Hong et al, 2005; Liu et al, 2014; Lee et al, 2015). Notably, CRMP2 was also found to be differentially expressed across different regions in the postmortem brain samples from schizophrenia patients (Edgar et al, 2000; Johnston-Wilson et al, 2000; Prabakaran et al, 2004; Beasley et al, 2006; Clark et al, 2006; Sivagnanasundaram et al, 2007; Martins-de-Souza et al, 2009a; Martins-de-Souza et al, 2009b; Martins-de-Souza et al, 2010). Dysregulation of the CRMP2 was also consistent in pharmacological and behavioral rodent models of schizophrenia (Paulson et al, 2004; Iwazaki et al, 2007; Lee et al, 2015). Furthermore, the brain-specific CRMP2-deficient mice displayed phenotypes reminiscent of schizophrenia patients (Zhang et al, 2016). These lines of evidence, thus, underscore the role of CRMP2 in etiopathogenesis of schizophrenia.

Because human-induced pluripotent stem (iPS) cell technologies have enabled an in vitro recapitulation of the neuropsychiatric disease pathogenesis (Balan et al, 2018), we set out to analyze iPS cells with *GLO1* disruption as a cellular model to uncover the missing link between enhanced carbonyl stress and the development of schizophrenia, at a very early developmental stage. To this end, we analyzed the effects of hyper-carbonyl (AGE) modification in iPS cells from a schizophrenia patient through multiple biological approaches. Our biochemical analysis identified CRMP2 as an important molecular target of AGE modification in iPS cells with enhanced carbonyl stress. X-ray crystallography and biochemical analyses revealed that the D-hook (dynamic binding

surface for dimerization) and T-site (tacking surface for tetramerization) of CRMP2, which are functionally critical for the reversible conformation required for CRMP2 activity, contribute to the transformative tetrameric complex of CRMP2. Strikingly, we also revealed that carbonylated CRMP2 (AGE-CRMP2) was stacked in the irreversibly formed multimer via AGE modifications at D-hook, T-site, and the outer surface of the CRMP2 complex, resulting in loss of the unique function of CRMP2 to bundle MTs. Collectively, the results obtained in the present study provide direct evidence showing the dysfunction of CRMP2 under enhanced carbonyl stress at the molecular and atomic levels and explains the cellular developmental deficits of iPS cells with GLO1 deficiency.

## Results

### iPS cells from a patient with schizophrenia carrying mutated *GLO1* and *GLO1*-knockout iPS cells exhibited impaired neurosphere growth under enhanced carbonyl stress

We previously reported that a *GLO1* frameshift mutation in a schizophrenia patient result in enhanced carbonyl stress with high plasma PEN levels (Toyosima et al, 2011). To uncover the fundamental link between *GLO1* mutation and schizophrenia pathogenesis under enhanced carbonyl stress, iPS cells derived from schizophrenia patients were precisely analyzed at the cellular and molecular levels. First, iPS cells derived from a schizophrenia patient with *GLO1* (fs) (SZ with *GLO1* frameshift mutation +/fs), iPS cells from a schizophrenia patient with normal *GLO1* (SZ with *GLO1* (+/+)), and iPS cells from a healthy control subject (control) (Fig S1) were cultured to induce the formation of neurospheres, which was observed 5 d after passage. Our neurospheres consisted almost entirely of neural stem or progenitor cells (Matsui et al, 2012). The number of neurospheres from SZ with *GLO1* (fs) iPS cells was 50% lower compared with those from control and SZ iPS cells ($P < 0.001$) (Fig 1A and B). To test whether enhanced carbonyl stress contributes to the observed growth deficit, the cells were treated with pyridoxamine, an RCC scavenger. Pyridoxamine treatment significantly increased the numbers of neurospheres from SZ with *GLO1* (fs) iPS cells ($P < 0.001$) (Fig 1A and B). To corroborate the effects of defects in *GLO1*, we introduced GLO1-deficient human iPS cells (*GLO1* (−/−) iPS cells) that were generated using the CRISPR/Cas9 system (Ohnishi et al, 2019). We confirmed that *GLO1* (−/−) iPS cells did not produce GLO1 protein in both neurospheres and differentiated neuronal cells by Western blotting (Fig 1C). The number of neurospheres from *GLO1* (−/−) iPS cells was 20% lower compared with that from *GLO1* (+/+) iPS cells ($P < 0.001$) (Fig 1D and E). The mean size of the neurospheres from *GLO1* (−/−) iPS cells (112 ± 31 $\mu$m) was reduced by 30% compared with that from *GLO1* (+/+) iPS cells (150 ± 41 $\mu$m) ($P < 0.001$) (Fig 1F). Pyridoxamine treatment significantly increased the number ($P < 0.01$) and size ($P < 0.05$) of neurospheres from *GLO1* (−/−) iPS cells (Fig 1E and F). We subsequently examined the neuronal cells that differentiated from *GLO1* (−/−) iPS cells. The *GLO1* (−/−) neuronal cells displayed significantly shortened neurites (38 ± 17 $\mu$m) when compared with the *GLO1* (+/+) neuronal cells (62 ± 13 $\mu$m) ($P < 0.001$) (Fig 1G and H).

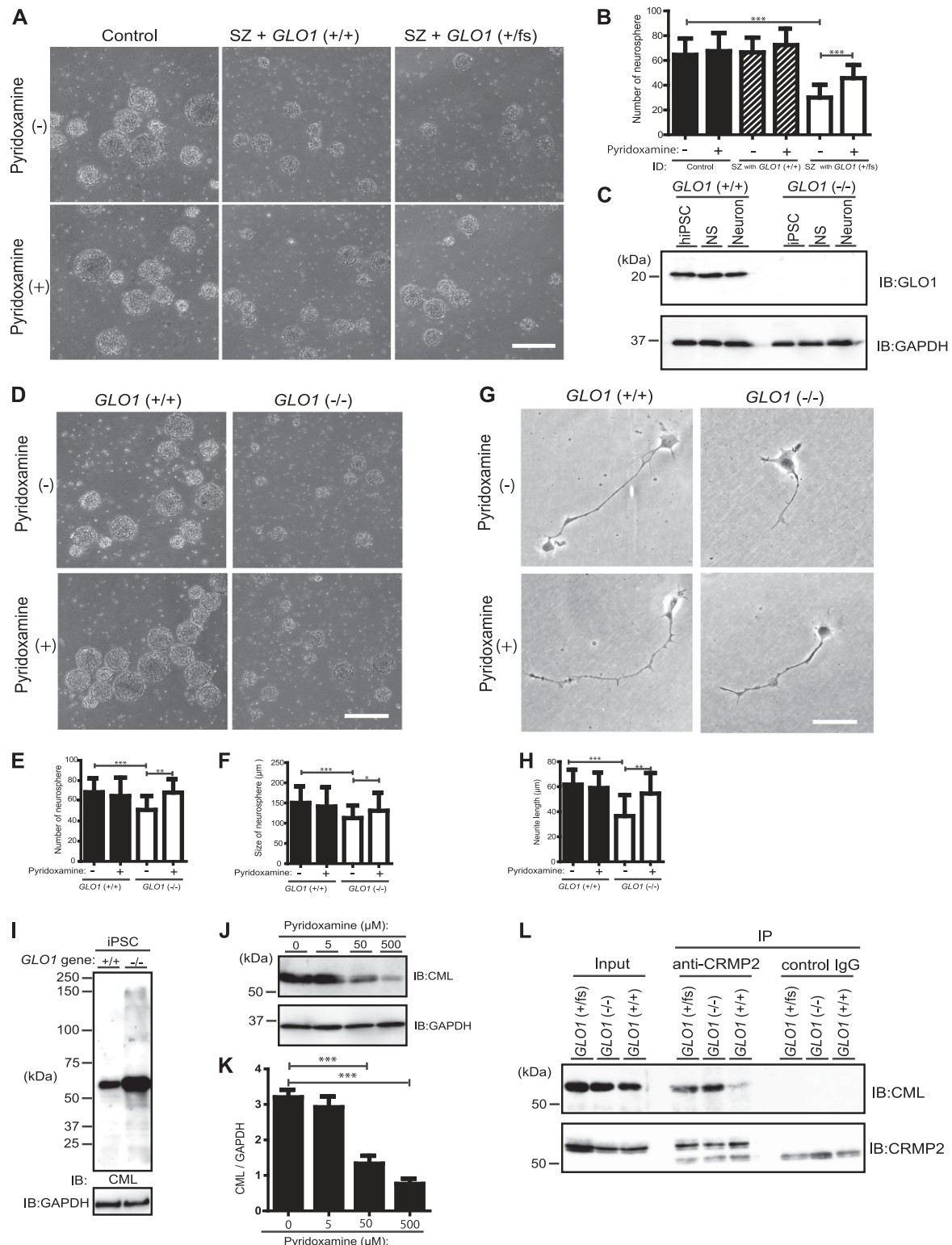

**Figure 1.  iPS cells from a schizophrenia patient with a *GLO1* mutation exhibited impaired neurosphere growth under enhanced carbonyl stress, and CRMP2 is a major target of AGE modification under enhanced carbonyl stress in *GLO1* (−/−) and *GLO1* (fs) iPS cells.**
**(A)** Bright-field images of neurospheres generated from iPS cells derived from a control individual, a schizophrenia (SZ) patient with wild type *GLO1* gene (SZ with *GLO1* (+/+)), and an SZ patient with the *GLO1* frameshift mutation (SZ with *GLO1* (fs)) (scale bar: 200 μm). **(B)** Quantification of the number of neurospheres. **(C)** Immunoblotting of GLO1 protein in *GLO1* (+/+) and *GLO1* (−/−) iPS cells. **(D)** Bright-field images of neurospheres that formed from *GLO1* (−/−) iPS cells in the presence or absence of pyridoxamine (scale bar: 200 μm). **(E, F)** Quantification of the size (E) and number (F) of neurospheres from *GLO1* (+/+) and *GLO1* (−/−) iPS cells. **(G)** Phase-contrast images

Similarly to the neurosphere-based findings, pyridoxamine treatment recovered the shortened neurite phenotype of GLO1 (−/−) neuronal cells (Fig 1H). We reasoned that the pronounced deficits in cellular phenotypes, observed in schizophrenia patient-derived iPS cells with GLO1 frameshift mutation might be contributed by the additional schizophrenia-associated genetic risk factors in the subject (Toyosima et al, 2011; Balan et al, 2014; Toyoshima et al, 2016). Taken together, these results indicate that the GLO1 loss of function (GLO1 (fs) mutation derived from the schizophrenia patient and the engineered GLO1 (−/−) mutation) impaired the cellular development of iPS cells because of the enhanced carbonyl stress.

### CRMP2 is a major target of AGE modification under enhanced carbonyl stress in GLO1 (fs) iPS cells derived from schizophrenia patients

Enhanced carbonyl stress leaves irreversible AGE footprints on proteins in cells, which often leads to loss of protein functions and could be a component of multiple cellular deficits and diseases. Therefore, we subsequently sought to identify the key target of the AGE modification in GLO1 (−/−) iPS cells. The AGE-modified proteins in GLO1 (−/−) iPS cells were immunoblotted using various types of anti-AGE antibodies that detect N-ε-(carboxymethyl) lysine (CML), N-ε-(carboxyethyl) lysine (CEL), methylglyoxal-derived hydroimidazolone (MG-H1), argpyrimidine (ARP), and PEN (Figs 1I and S1B). Notably, a strong band corresponding to ~60 kD was identified in GLO1 (−/−) iPS cells using CML antibody (Fig 1I). This increased CML modification was also observed in the GLO1 (−/−) neurospheres (Fig S1B). The CML modification of the ~60-kD band was reduced by the pyridoxamine treatment in a dose-dependent manner (Fig 1J and K), which indicated that the ~60-kD proteins are the major AGE-modified components under enhanced carbonyl stress and could be suppressed by pyridoxamine treatment.

To identify the major AGE-modified proteins under enhanced carbonyl stress, the AGE-modified proteins were enriched from cultured GLO1 (−/−) iPS cells by anion-exchange chromatography (Mono Q; GE Healthcare), and the fraction containing the highly CML-modified 60-kD proteins was identified by immunoblotting with anti-CML antibody. After the fraction was developed on a conventional SDS–PAGE gel, the major band at 60 kD was analyzed using an liquid chromatography-mass spectrometry/mass spectrometry (LC-MS/MS) system (Thermo Fisher Scientific) and identified as CRMP2 (Fig S1C and D). To confirm the result, CRMP2 was immunoprecipitated by anti-CRMP2 antibody and detected as the exact CML-modified protein in the GLO1 (−/−) and GLO1 (fs) iPS cells using anti-CML antibody (Fig 1L). Considering the essential role of CRMP2 in regulating the cytoskeletal dynamics in relation to the neural development (Arimura et al, 2000; Cole et al, 2004; Quach

et al, 2004; Kawano et al, 2005), we revealed that the impaired cellular and developmental phenotypes manifested in the neural cells derived from GLO1-deficient iPS cells were stemmed mainly from the carbonylation (CML) of CRMP2. However, the role of other carbonylated proteins contributing to these observed phenotypes in a synergistic manner can also not be excluded. Because the role of CRMP2 in schizophrenia has been evidenced from the patient studies, we reasoned that the AGE modification of the CRMP2, resulting from schizophrenia-associated genetic aberrations, in neural progenitor cells can contribute to the downstream neurodevelopmental deficits relevant for the schizophrenia pathogenesis, at least in a subset of patients.

### Sites of AGE modification are widely distributed in the functional motifs of CRMP2

Because several functional regions and numerous PTMs of CRMP2 reportedly contribute to its function and regulation, we then sought to determine the exact position of carbonylation sites (AGE sites) and investigate the effects of AGE modification on CRMP2 function. To identify the AGE sites of CRMP2, CRMP2 purified from GLO1 (−/−) iPS cells and recombinant AGE-CRMP2 proteins were precisely analyzed through advanced mass spectrometric approaches (Fig 2A). CRMP2 and AGE-modified proteins enriched from GLO1 (−/−) iPS cells by anion-exchange chromatography were further purified by high-resolution size-exclusion chromatography (HiRes-SEC) (Ogawa & Hirokawa, 2018), and the eluate containing CRMP2 was confirmed using anti-CRMP2 antibody. Recombinant human CRMP2 (C532, residue 1–532) was bacterially expressed, purified, and incubated for in vitro carbonylation in the presence of glyoxal as RCC (Fig S1E). The identification of AGE sites at the peptide level is generally challenging because the loss of positive charges by AGE modification reduces the ionization of peptides to be detected by mass spectrometry. In addition, aldehyde groups might react with primary amines during the sample preparation process, which hampers the subsequent analysis. Therefore, we performed chemical derivatization to ensure AGE sites intact, before the peptide mapping using mass spectrometry. The AGE sites were derivatized with Girard's Reagent T (GRT) (Yang et al, 2014) to induce the formation of hydrazone products in which the Schiff base was subsequently reduced by $NaBH_4$. The derivatized protein was digested and analyzed using an LC-MS/MS system. Using our strategy, mass analyses covered 96.68% of endogenous CRMP2 in GLO1 (−/−) iPS cells and 100% of recombinant CRMP2 (C532) in terms of protein sequence coverage, respectively (Figs 2A and S2A).

29 sites overlapped between the 50 sites in CRMP2 from the GLO1 (−/−) iPS cells and 62 sites in in vitro expressed CRMP2 (7 Arg, 11 Lys, 5 Pro, and 6 Thr), and 12 and 14 other sites in the former and latter proteins, respectively, were very closely localized to each other

---

of neurons derived from GLO1 (+/+) and GLO1 (−/−) iPS cells in the presence or absence of pyridoxamine (scale bar: 25 $\mu$m). **(H)** Quantification of the neurite length from images shown in (G) (statistics: data are presented as the means ± SDs; *$P$ < 0.05, **$P$ < 0.01, and ***$P$ < 0.001, as determined by two-way ANOVA followed by Tukey's multiple-comparisons test; (B), (E), and (F): n = 60; (H): n = 30). **(I)** Immunoblotting of AGE-modified proteins in GLO1 (−/−) iPS cells by anti-AGE antibodies that detect N-ε-(carboxymethyl) lysine (CML). **(J)** Immunoblotting of the CML modification at ~60 kD was gradually decreased by pyridoxamine treatment in a dose-dependent manner. **(K)** Quantification of CML modification at ~60 kD after pyridoxamine treatment (statistics: data are presented as the means ± SDs; **$P$ < 0.01 and ***$P$ < 0.001, as determined by one-way ANOVA followed Tukey's multiple comparison test; n = 9). **(L)** CML-modification of CRMP2 immunoprecipitated by anti-CRMP2 antibody in GLO1 (−/−) and GLO1 (fs) iPS cells. IB, immunoblot; IP, immunoprecipitate.
Source data are available for this figure.

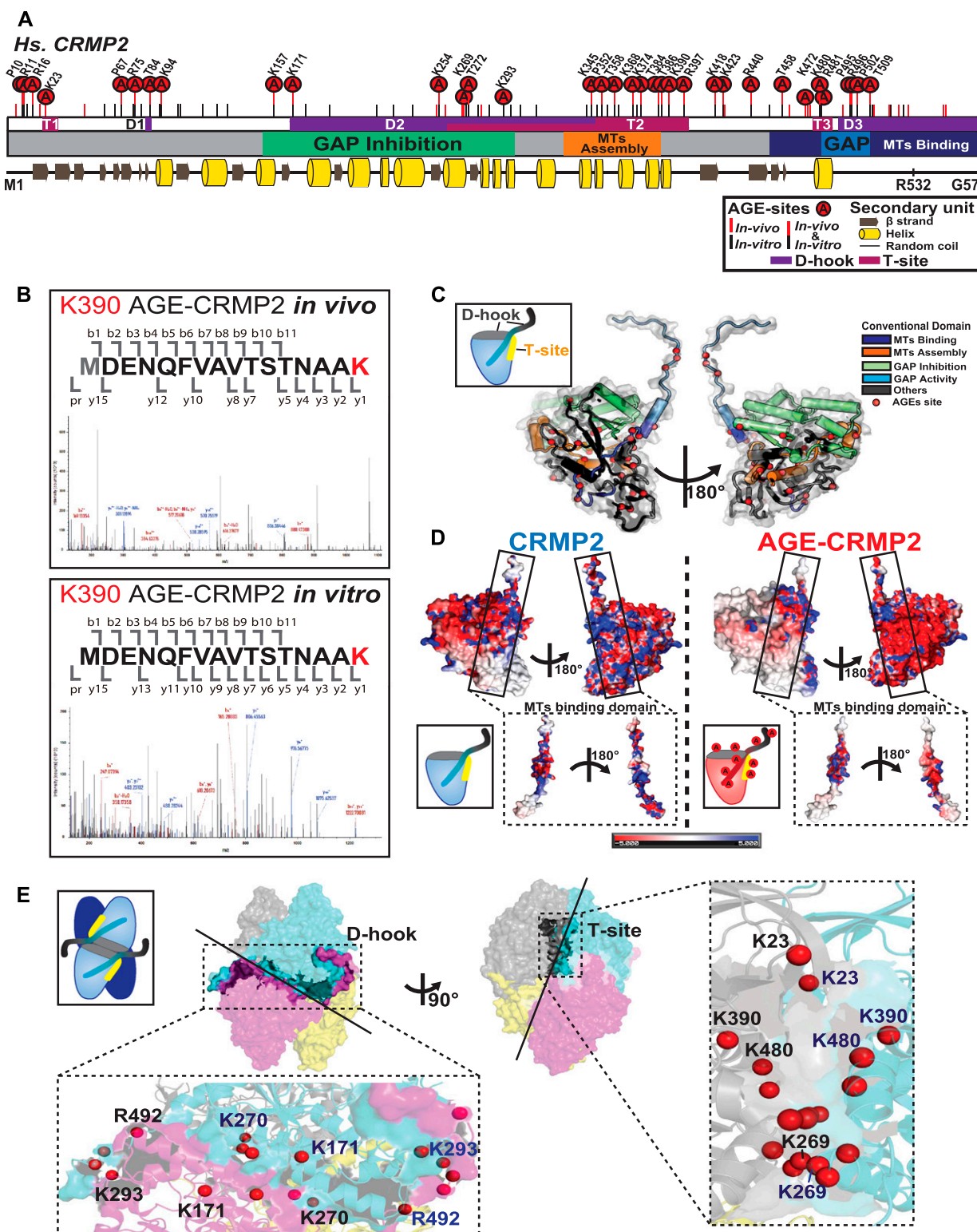

**Figure 2. AGE sites are widely distributed around conventional functional domains and located in two structural interfaces of the functional tetramer complex, D-hook and T-site.**

**(A)** Distribution of AGE sites in CRMP2 in vivo and in vitro. 29 (7 Arg, 11 Lys, 5 Pro, and 6 Thr) overlapped (indicated by "A" in a red circle) among the 50 sites in in vivo CRMP2 C532 (showed with a red line) and 62 sites in in vitro CRMP2 C532 (shown with a black line). The conventional motifs identified in previous studies related to MT assembly are shown at the top, and the 2D structural units are shown at the bottom. **(B)** An AGE site on K390 was determined with high accuracy based on the MS/MS spectra obtained both in vivo and in vitro. **(C)** AGE sites are distributed on the outer surface around the conventional functional domains in the CRMP2 structure: GAP activity (6 sites),

within ±5 amino acids (Figs 2A and S2B), which suggest that the in vitro glyoxal-treated CRMP2 highly resembles the in vivo AGE-modified CRMP2 in terms of the distribution of AGE sites and thereby is appropriate to be used for further in vitro analysis. In particular, the AGE site on K390 was determined with high accuracy by the MS/MS spectrum in CRMP2 from both systems (Fig 2B). The AGE sites were widely distributed on the outer surface around conventional domains that were previously reported to regulate MT assembly: GAP activity (6 sites), inhibition of GAP activity (19 sites), promotion of MT assembly (10 sites), and interaction with MTs (21 sites) (Fig 2A and C). Because the surface potential of a protein is essential for its activity, we subsequently simulated the mechanism underlying the effects of AGE modification on surface charges by mutating the determined AGE-modified Lys/Arg residues to their derivative form, CML/GOARG, on the CRMP2 structure in silico (Fig 2C). Compared with the wild-type protein, simulated AGE-CRMP2 suffers drastic changes in terms of its electrostatic distribution and exhibits a great loss of positive charges (Fig 2D, in blue) and an apparent increase in negative charges (Fig 2D, in red) because of the AGE modification that converts positively charged Arg and Lys to uncharged and negatively charged derivatives. This AGE-induced reorganization of electrostatic surface would affect the function of CRMP2.

### AGE sites are located in D-hook and T-site, which are central interfaces for the functional tetramer complex

In addition to the wide distribution of functional motifs, 42 AGE sites were found to be involved in the protein assembly interface of CRMP2, which is critical for the organization of the active functional unit of the tetramer CRMP2 complex (Zheng et al, 2018) (Figs 2E and S2B). To facilitate our understanding of the functional link between the structure and AGE modification of CRMP2, we focused on two structural interfaces: the D-hook (dynamic binding surface for dimerization) and the T-site (tacking surface for tetramerization). Our structural analysis of AGE-CRMP2 by X-ray crystallography revealed that D-hook has an extended conformation composed of a basal N-terminal β8 sheet, an intermediate region from β11 to H14 and a flexible C-terminal loop from A489 to the end. The T-site is composed of three structural interfaces: rigid N-terminal β1–β2 sheets, an intermediate surface from H8 helix to β15 sheet and a C-terminal H18 helix (Figs 2A and E, and S2B). The functional tetrameric complex of CRMP2 is predicted to be assembled through successive two-step folding via the D-hook and T-site. In AGE-CRMP2, we identified 18 AGE sites (average of nine sites in each monomer) within both the D-hook and T-site, including K23, K269, K390, and K480 in the T-site (Fig 2E). Considering the structural contribution of D-hook and T-site to the formation of the tetrameric complex, AGE modifications should directly affect the assembly dynamics and activity of CRMP2. Therefore, we conducted further

analyses on the function and structure of CRMP2 under AGE modification.

### AGE modification diminishes the MT-bundling activity of CRMP2

Because CRMP2 reportedly promotes MT elongation and bundling, we evaluated the in vitro activity of AGE-CRMP2 in comparison with that of unmodified CRMP2 using a total internal reflection fluorescence (TIRF) microscope. Polymerized MTs were incubated for 5 min in the presence of unmodified CRMP2 protein C532 or AGE-modified CRMP2 C532, and the length and intensity of the MT filaments were measured. Unmodified CRMP2 elicited significant bundling of MTs with increased length and intensity. In contrast, AGE-CRMP2 did not increase the length and intensity of MTs, which indicated that AGE modification disturbed the MT-bundling activity of CRMP2 (Fig 3A and B; filament length: 21.23 ± 3.13, 6.96 ± 0.54, and 9.43 ± 0.87 for MTs + CRMP2, MTs + AGE-CRMP2, and MTs + BSA, respectively; filament intensity: 1,536.24 ± 154.93, 352.74 ± 11.22, and 349.25 ± 10.93 for MTs + CRMP2, MTs + AGEs-CRMP2 and MTs + BSA, respectively). The MT-bundling and MT-binding activities were further investigated through a conventional centrifugal analysis. Specifically, the MT-bundling activity of CRMP2 and AGE-CRMP2 was examined by low-speed centrifugation (6,000$g$, 10 min) (Fig 3C). The number of bundled MTs was increased by unmodified CRMP2 in a dose-dependent manner, whereas AGE-CRMP2 did not show MT-bundling activity (Fig 3C and D, upper panel). Unmodified CRMP2 was co-sedimented with bundled MTs in pellets, whereas AGE-CRMP2 existed in the supernatant (Fig 3C and D, lower panel). Interestingly, we found that AGE-CRMP2 was stacked in irreversible polymerized forms that can be detected as upper bands on SDS–PAGE gel (Fig 3C). Furthermore, we analyzed the affinity of CRMP2 with polymerized MTs through a conventional high-speed MTs co-sedimentation assay (76,000$g$, 30 min). Unmodified CRMP2 displayed high affinity to polymerized MTs in a dose-dependent manner; however, AGE-CRMP2 showed reduced affinity to polymerized MTs with irreversibly polymerized upper bands on the SDS–PAGE gel (Fig S3A and B). Different activities between unmodified CRMP2 and AGE-CRMP2 were also confirmed through an assay of MT turbidity that measures the degree of polymerization, elongation, and bundling of MTs (Fig S3C). Compared with the moderate promotion of MT polymerization by unmodified CRMP2, AGE-CRMP2 did not promote MT assembly well, resulting in a growth rate that was similar to that of the negative control (tubulin only) (Fig S3C). These results clearly show that AGE modification impairs the function of CRMP2, and in particular, this modification dampens the MT-bundling activity of CRMP2 in vitro. The effect of AGE modification on the MT-bundling activity of CRMP2 was also evaluated in the GLO1 (+/+) and GLO1 (−/−) iPS cells with/without glyoxal. To quantify the stability of MTs, iPS cells were treated by

---

inhibition of GAP activity (19 sites), promotion of MT assembly (10 sites), and interaction with MTs (21 sites). The structure model (PDB code: 5MKV) was used for representation here. **(D)** Simulation of surface potential changes of CRMP2, which showed that AGE modification induced a great loss of positive charges (in blue) and an apparent increase in negative charges (in red). The MT-binding domain is highlighted. The structure models (PDB codes: 6JV9 and 6JVB) were used for simulation analysis. **(E)** AGE sites were localized in two structural interfaces associated with the self-assembly of CRMP2: D-hook and T-site. D-hook comprises 10 and 8 AGE sites for molecule A (in magenta) and the adjacent molecule B (light blue). In total, 18 AGE sites of D-hook are shown in the bottom-left panel. T-site contains 9 AGE sites for each molecule. In total, 18 AGE sites of T-site are shown in the right panel. Representative AGE-modified residues are labelled.

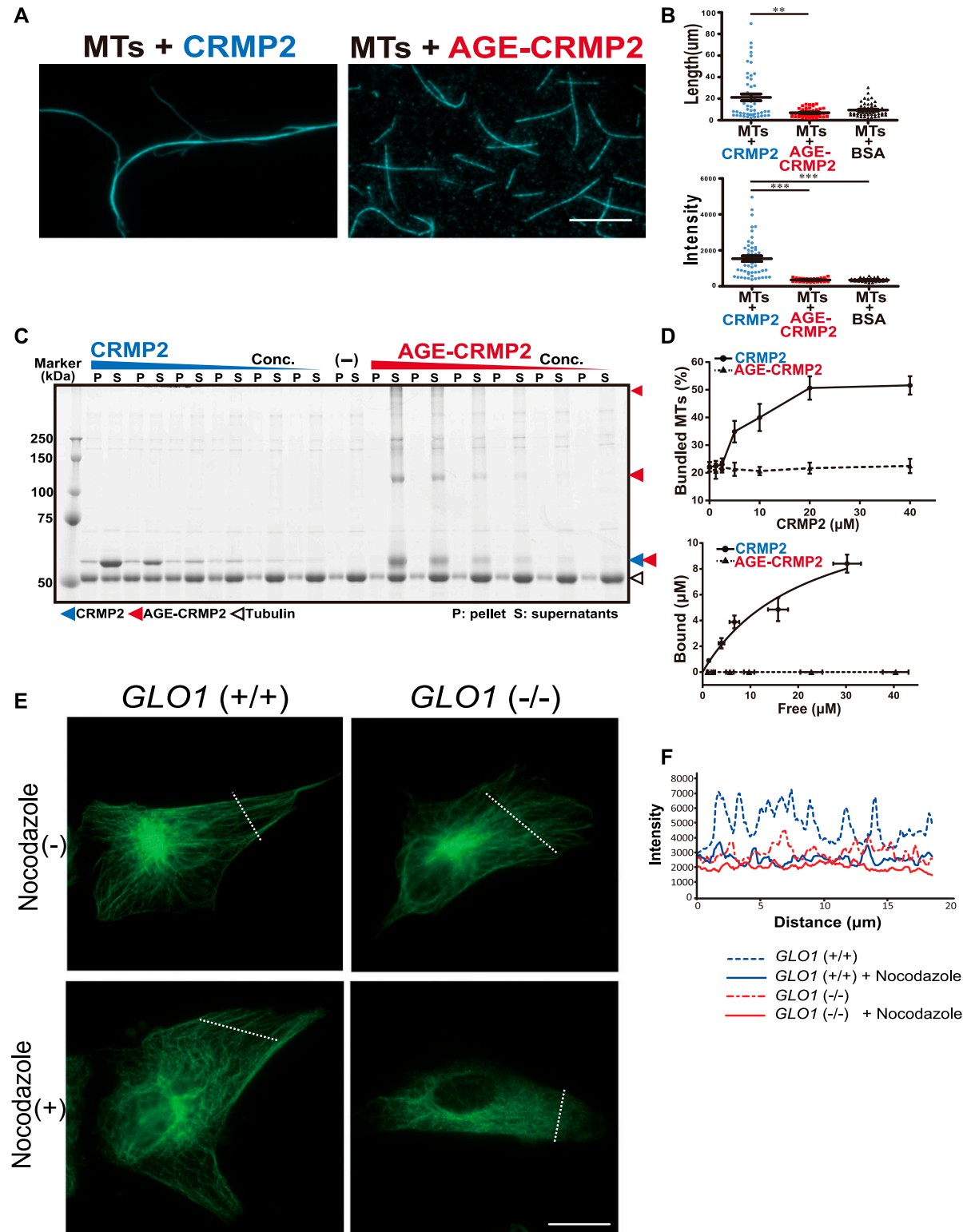

**Figure 3. AGE modification diminishes the MT-bundling activity of CRMP2.**
**(A)** CRMP2 C532 protein significantly bundled polymerized MTs (protein ratio = 1 μM tubulin dimer: 4 μM CRMP2, left panel). In contrast, AGE-CRMP2 did not display MT-bundling activity (protein ratio = 1 μM tubulin dimer: 4 μM AGE-CRMP2, right panel) (scale bar: 10 μm). **(B)** Quantification of the length and intensity of MTs (statistics: data are presented as the means ± SEMs; **$P < 0.01$ and ***$P < 0.001$, as determined by the Kruskal–Wallis with Dunn's multiple comparison test; n = 50). **(C)** Low-speed MT co-sedimentation assay for assessing the bundling activity of CRMP2 (or AGE-CRMP2) with polymerized MTs. Various concentrations of proteins were mixed with Taxol-stabilized MTs and then subjected to co-sedimentation (6,000$g$, 10 min) assessment, and the bundled MTs were then sedimented to obtain a pellet (P). The P and

nocodazole (a MT-destabilizing drug) for 15 min, fixed, and stained, and the survived stable MTs were monitored (Figs 3E and F and S3D and E). The *GLO1* (+/+) iPS cells displayed stable MTs with strong fluorescence intensity even under nocodazole treatment. In contrast, the *GLO1* (−/−) iPS cells displayed destabilized tubulin after nocodazole treatment, as demonstrated by a blurred fluorescence intensity, which indicated that the MTs of *GLO1* (−/−) iPS cells were less stable than those of the *GLO1* (+/+) iPS cells (Fig 3E and F). After the elicitation of carbonyl stress by glyoxal treatment, the MTs of *GLO1* (+/+) iPS cells displayed similar stability to those of *GLO1* (−/−) iPS cells (Fig S3D and E). These results suggest that the AGE modification of CRMP2 disrupts the MT-bundling ability of this protein in cellular system.

## AGE modification induces the formation of irreversible di-Lys bonds

Next, we asked how AGE modification elicits impairment of CRMP2 activity. To this end, the structures of unmodified CRMP2 (C532) and AGE-CRMP2 were analyzed by X-ray crystallography at 2.26 and 2.00 Å resolutions, respectively (Fig 4). By comparing the overall tetrameric complex structures, apparent structural differences were observed in the N-terminal region, especially for the β2 and β3 sheets that include the AGE target K23 in T-sites. In AGE structure, β2 sheet underwent a sharp inclination and β3 sheet broke into two separated β strands with a linking loop (Fig 4A). Based on the density maps of unmodified and AGE-CRMP2 structures, much significant difference was observed within T-site. In the AGE-CRMP2 complex structure, three Lys pairs (K23–K480, K269–K269, and K390–K390) of T-site were AGE-modified in close distance of the ε-ammonium group with extra electron density in Fo-Fc map (compared with unmodified CRMP2 structure) (Fig 4B). Furthermore, the electron density of the AGE modification was also observed at K43, K56, and K451 on the outer surface of tetrameric complex with their AGE-modified Lys side chains in outward protrusions (Fig 4B). Because AGE-derivative residues comprise not only the intramolecular type such as CML but also the intermolecular types such as GOLD (glyoxal pathway), DOLD (3DG pathway), and MOLD (methylglyoxal pathway), the extra electron density indicates the formation of irreversible di-Lys bonds within/between the CRMP2 tetramer complexes by AGE modification (Fig 4B).

## AGE modification induces CRMP2 dysfunction through the irreversible formation of a multimer complex

Because previous studies have indicated that CRMP2 potentially forms dynamic transformative conformations between its tetramer (Zheng et al, 2018) and monomer (or dimer) (Inagaki et al, 2001;

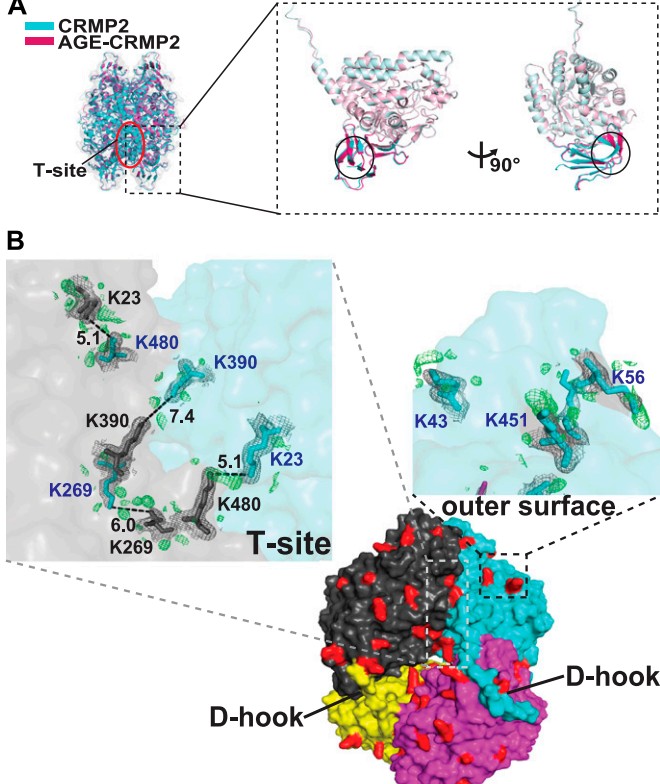

**Figure 4. AGE modification induces the formation of irreversible di-Lys bonds on T-site and the outer surface of the tetramer complex.**
**(A)** Superposition of the tetrameric complexed structures of CRMP2 (C532, in blue) and AGE-CRMP2 (in red) solved by X-ray crystallography at 2.26- and 2.00-Å resolution, respectively. In the AGE-modified structure, the β2 sheet underwent a sharp inclination, and the β3 sheet broke into two separated β strands with a linking loop. **(B)** Comparison of the electron density maps of unmodified and AGE-modified CRMP2 structures. The additional electron density in the Fo-Fc map (AGE-Lys, in red-coloured density) was observed within the T-site and on the outer surface of the tetrameric complex. In the AGE–CRMP2 complex structure, three Lys pairs (K23–K480, K269–K269, and K390–K390, in the upper left panel) within the T-site and in close proximity to the ε-ammonium group were modified by AGE to likely form di-Lys bonds. Three Lys residues (K43, K56, and K451, in the upper right panel) were modified by AGE, and their modified side chains were located in outward protrusions. The 2Fo-Fc (in grey) and Fo-Fc (in green) density maps were contoured at 1σ and 3σ, respectively.

Fukata et al, 2002; Charrier et al, 2003; Cole et al, 2004; Quach et al, 2004; Niwa et al, 2017) forms to generate its specific function and regulation, we further evaluated the functional states of unmodified and AGE-modified CRMP2 proteins by measuring the flexibility and the dynamic size of the complexes in solution. To investigate the flexibility and the stability of CRMP2 in solution,

---

supernatant (S) fractions were used to assay the MT-bundling and MT-binding activities of unmodified and AGE-modified CRMP2 to bundled MTs. The irreversible polymers of AGE-CRMP2 were also observed as the upper bands on an SDS–PAGE gel. **(D)** Quantitative analysis of the low-speed co-sedimentation assay results. Three replicates of the co-sedimentation data were used for the quantification. The MTs in the P and S fractions were quantified to analyze the proportion of bundled MTs (upper panel), and the amounts of bound proteins were plotted as functions of the amounts of unbound proteins (lower panel; the bar graph represents the means ± SDs). **(E)** Assay of the stability of MTs in *GLO1+/+* and *GLO1−/−* iPS cells in the presence of nocodazole, an MT-destabilizing drug (scale bar: 25 μm). **(F)** Comparison of the fluorescence intensity of stabilized MTs. The fluorescence intensity of MTs was measured at a distance equal to one-third of the distance from the leading edge of the cell to the edge of the nucleus.
Source data are available for this figure.

unmodified and AGE-modified CRMP2 proteins were analyzed by differential scanning calorimetry (DSC). The DSC results showed that the thermal transition midpoint (Tm) of AGE-CRMP2 rose to 64.4°C (+1.8°C) compared with that of CRMP2 (62.6°C), which indicated that AGE-CRMP2 gained higher thermal stability and lower flexibility after its transformation (Fig 5A). To precisely evaluate the dynamic states of CRMP2 after AGE modification, unmodified and AGE-modified recombinant CRMP2 proteins were centrifuged to separate the large aggregates, and the supernatants were analyzed through a HiRes-SEC–based assay (Ogawa et al, 2017). On the SEC chromatogram, unmodified CRMP2 exhibited a major tetramer complex and two minor peaks that represent its octamer and monomer forms, which reflected a dynamic equilibrium between reversibly formed complexes in solution (Fig 5B). In contrast, AGE-CRMP2 was stacked in a multimer form, which indicated that the transformative dynamics was disrupted by the irreversible AGE modification. In addition, on SDS–PAGE, AGE-CRMP2 displayed multiple upper band shifts with substantially higher molecular weights than 60 kD, which indicated that AGE modification induced irreversible cross-linking within/between the complexes that cannot be broken even under strong reducing and unfolding conditions (Fig 5B).

Furthermore, whether the irreversible cross-linking of CRMP2 under enhanced carbonyl stress was produced in iPS cells was also examined. To exclude the bias in the epitope recognition of anti-CRMP2 antibody against AGE-CRMP2, N-terminal Myc-tagged CRMP2 was expressed in iPS cells and analyzed using anti-Myc antibody. In the GLO1 (−/−) iPS cells, Myc-CRMP2 displayed multiple upper band shifts above 60 kD probably because of irreversible cross-linking, which is consistent with our in vitro results. These protein phenotypes of CRMP2 in GLO1 (−/−) iPS cells were suppressed by pyridoxamine treatment (Fig 5C and D). Collectively, the results demonstrate that AGE modification causes irreversible polymerization and aggregation (both CRMP2 from the GLO1 (−/−) iPS cells and in vitro expressed CRMP2) as well as the formation of intermolecular bonds within/between the tetramers, which disturbs the transformative dynamics of CRMP2. This molecular behavior of the multimerized AGE–CRMP2 complex in solution is consistent with the rigid cross-linking of di-Lys bonds both within the tetramer complex and between the complexes suggested by the electron density maps obtained from the crystal structure of AGE-CRMP2 (Fig 4B).

Based on all the above-described evidence, we propose a new model for the molecular pathogenesis in a subset of schizophrenia under enhanced carbonyl stress (Fig 6). Under healthy conditions, CRMP2 proteins exist as a tetrameric complex in dynamic equilibrium between its transformative units, such as octamer, tetramer, and monomer (Fig 6A), and in this state, CRMP2 showed high MT-bundling activity (Fig 6B). In contrast, under enhanced carbonyl stress, CRMP2 proteins are carbonylated as a major target of AGE modification, and these effects elicit the multimerization and aggregation of CRMP2 through the formation of irreversible di-Lys bonds within/between the tetrameric complexes (Fig 6C). This damage to the transformative dynamics of CRMP2 dampens the MT-bundling activity of CRMP2, resulting in abnormal cellular deficits and potentially leading to schizophrenia susceptibility (Fig 6D).

## Discussion

Enhanced carbonyl stress leaves irreversible footprints on its target molecules in biological systems. In this study, we found the direct effects of enhanced carbonyl stress on CRMP2 via AGE modification. AGE sites were found to be distributed in functional motifs of CRMP2, particularly D-hook and T-site, which are critical for the formation of the active CRMP2 tetramer. Here, we analyzed the pathological mechanism of AGE modification on CRMP2 activity from two aspects. First, AGE modification altered the normal charge distribution in multiple functional regions of CRMP2 (Fig 2A, C, and D), which directly interfered with the CRMP2–MT or CRMP2–tubulin interactions (Figs 3C and D, and S3A–C). Second, AGE modification surprisingly produced covalent di-Lys linkages within the T-site of the CRMP2 tetramer (Fig 4B). Moreover, AGE-modified sites were also located on the outer surface of the tetrameric CRMP2 complex, and their AGE-modified Lys side chains were located in outward protrusions, which resulted in the production of irreversible bonds between the complexes and, therefore, in the multimerization and aggregation of CRMP2. Taking into consideration that the transformative conformation is a critical factor for CRMP2 activity, we conclude that the loss of this transformative conformation in the multimer and aggregates via the irreversible AGE modification is the fundamental molecular pathology of CRMP2 dysfunction under enhanced carbonyl stress.

In this article, we analyzed the AGE sites of CRMP2 both in iPS cells and in vitro. By using LC/MS analysis, we qualitatively determined more than 60 AGE sites. The quantitation of stoichiometry distribution of each AGE site in iPS cells and post-mortem brain should be further investigated in future. Based on the comparison of our data in iPS cells (Fig 5C and D) and in vitro (Fig S1E), we speculate that there is a similar stoichiometry of AGE modification between our iPS cell—derived and in vitro glyoxal-induced CRMP2 proteins.

It has been reported that monomeric CRMP2 promotes MT assembly when the soluble GTP-tubulin dimer breaks the stable tetrameric CRMP2 to form a complex composed of one CRMP2 monomer and the tubulin dimer (Fukata et al, 2002; Niwa et al, 2017). Moreover, tetrameric CRMP2 was recently reported to bundle MTs, and this activity decreased after the CRMP2 tetramer was decomposed into dimers through phosphorylation (Zheng et al, 2018). Therefore, it could be analogized that each specific function of CRMP2 corresponds to a specific conformation. Our HiRes-SEC assay showed that soluble free CRMP2 mainly exists as tetramer in dynamic equilibrium with its monomer and octamer forms (Fig 5D). When this dynamic transformation was stacked by AGE modifications (Fig 5D), both MT-assembly and MT-bundling activities were irreversibly lost (Fig 3). Taken together, our observations show that the dynamic transformative conformation is fundamental for the activity of CRMP2 against its substrate in the specific situation.

A number of PTMs were previously reported to regulate multifaceted functions of CRMP2. For instance, the functions of CRMP2 is reportedly promoted by SUMOylation and inhibited by phosphorylation via GSK3β, CDK5, Rho, and Fer kinases (Arimura et al, 2000; Yoshimura et al, 2005; Cole et al, 2007). However, the molecular and structural mechanism through which these modifications control

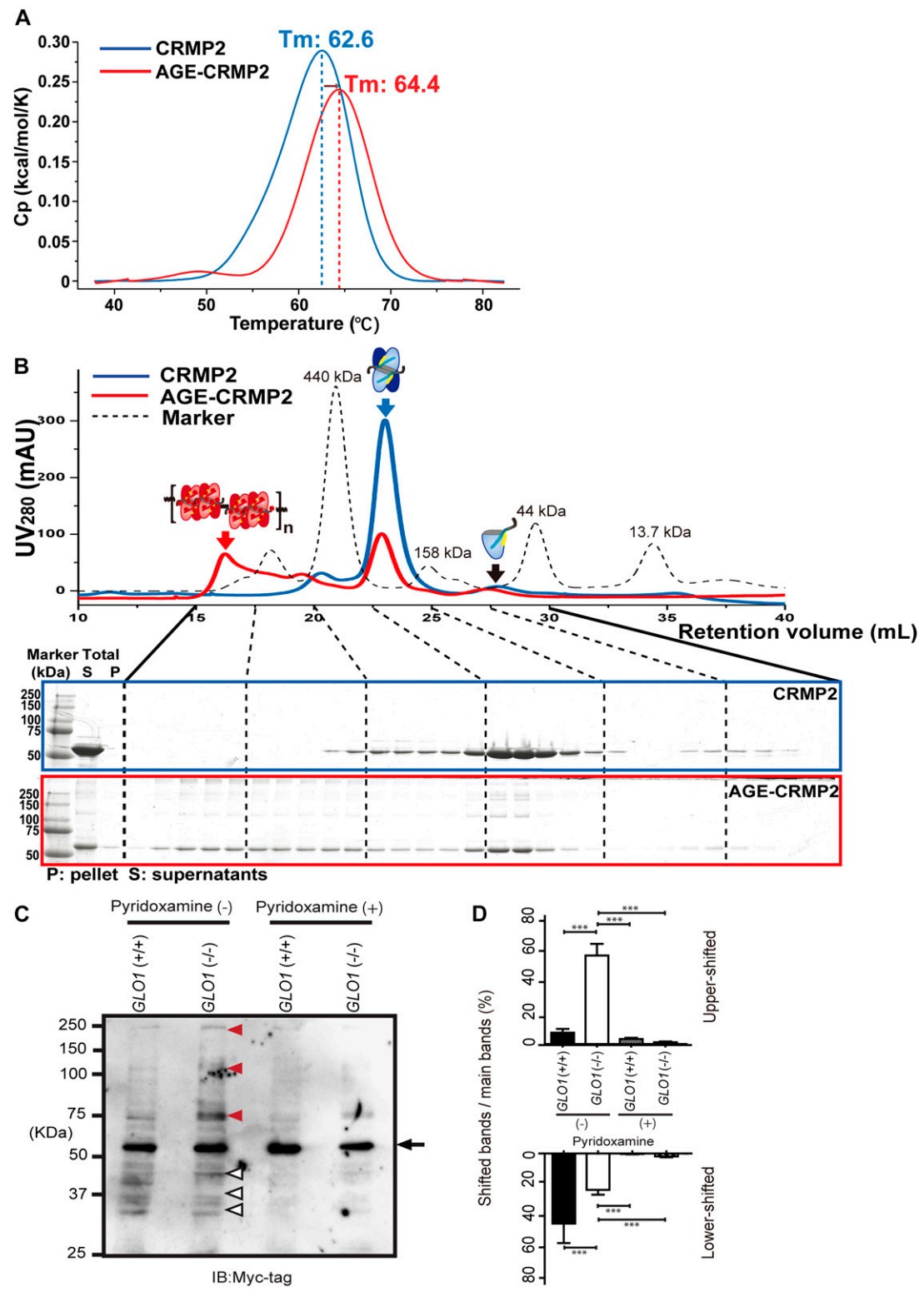

**Figure 5. AGE modification induces CRMP2 dysfunction through formation of an irreversible inactive conformation.**
**(A)** DSC analysis of unmodified and AGE-modified CRMP2 proteins to evaluate the flexibility and dynamic size of the complexes in solution. The thermal transition midpoint (Tm) of AGE-CRMP2 (in red) increased by 1.8°C–64.4°C from that of CRMP2 (62.6°C). **(B)** High-resolution size exclusion chromatography (HiRes-SEC) analysis of unmodified CRMP2 (in blue) and AGE-CRMP2 (in red) proteins to evaluate the dynamic size of the complexes in solution. The chromatogram of a standard mixture of molecular weight marker proteins is displayed with a black dotted line (ferritin, 440 kD; aldolase, 158 kD; ovalbumin, 44 kD; and ribonuclease A, 13.7 kD). The irreversibility of the complex in each fraction was monitored on the SDS–PAGE with a strong reducing environment (CRMP2 in a blue box, AGE-CRMP2 in a red box). The irreversible

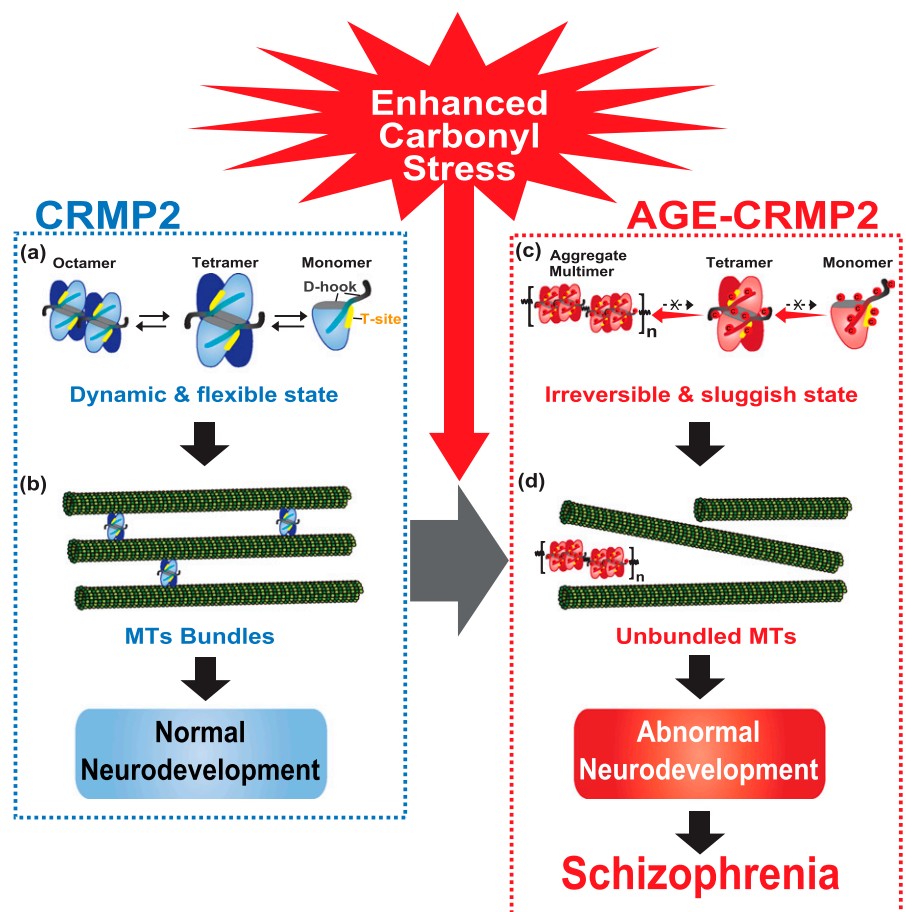

**Figure 6. Model of the molecular pathogenesis of a subset of schizophrenia under enhanced carbonyl stress.**
**(A)** Under healthy conditions, CRMP2 proteins exist as a tetrameric complex in dynamic equilibrium between its transformative units, such as its octamer, tetramer, and monomer forms. **(B)** In this state, CRMP2 showed high MT-bundling activity. **(C)** Under enhanced carbonyl stress, CRMP2 proteins are hyper-carbonylated as the major target of AGE modification, and this effect induces the irreversible polymerization and aggregation of CRMP2 through the formation of di-Lys bonds within and between the tetrameric complexes. **(D)** Loss of the transformative dynamics of CRMP2 diminishes the MT-bundling activity of CRMP2, which results in abnormal cellular and developmental deficits, and this effect constitutes the pathogenic basis of a schizophrenia subset under enhanced carbonyl stress.

CRMP2 function remained largely elusive. Interestingly, most of the reported phosphorylation sites are located within the D-hook and T-site regions. For example, Fer phosphorylates Y479 and Y499 within T-site and D-hook of CRMP2 and inhibits MT-bundling activity by demolishing the tetramer complex. Based on our observation of the dynamic CRMP2 transformation, these PTMs might tune the activity of CRMP2 by "reversibly" changing its functional tetramer states. In addition, many PTM sites are also overlapped or closely situated with the AGE sites, such as N6-succinyllysine at K258, SUMOylation at K374, phosphorylation at K514, and acetylation at K525 (Fig 2); therefore, AGE modification might also disrupt the normal regulation system of CRMP2 by destroying the target sites or the motif sequence targets of PTMs.

In this study, we revealed cellular deficits in neurosphere formation under carbonyl stress in iPS cells from a schizophrenia patient with the *GLO1* frameshift mutation. The phenotype of the differentiated neurons from *GLO1* (−/−) iPS cells, which showed a shortened

neurite length, highly resembles the previously reported phenotypes of iPS cells in schizophrenia (Brennand et al, 2011; Robicsek et al, 2013; Toyoshima et al, 2016). Furthermore, we identified CRMP2 as a major target of AGE modification in *GLO1* (−/−) iPS cells. Other proteins have been reported as targets of AGE modification, such as PEN-modified albumin in plasma (Koike et al, 2015) and ARP-modified SBP1 (selenium binding protein 1) (Ishida et al, 2017) in blood of schizophrenic patients. The targets of AGE modification might differ depending on various factors, such as cell type, organs, developmental stages, and pathogenic contexts. Not only cytoskeletal proteins, such as CRMP2, but also some specific receptors, channels, kinases, and phosphatases can be affected by AGE modification. In addition to carbonyl stress, other pathogenic factors can affect CRMP2 activity. Transcriptional and translational defects might be involved in the loss of CRMP2 activity observed in psychiatric disorders. The defect of upstream regulation in pathogenic contexts might also affect CRMP2

polymers of AGE-CRMP2 were observed as the upper bands on the SDS–PAGE gel (shown in a red box). The supernatants (S, samples for HiRes-SEC) and pellets (P) of the total protein samples were also analyzed by SDS–PAGE (shown at the far left of the SDS–PAGE images) to assess the AGE-induced aggregation. **(C)** Immunoblotting of transfected Myc-CRMP2 in *GLO1* (+/+) and *GLO1* (−/−) iPS cells with or without pyridoxamine treatment. The normal bands at 60 kD (black arrow), upper shifted bands (red arrow heads), and lower shifted bands (white arrow heads) of CRMP2 are indicated. **(D)** Comparison with upper or lower shifted bands to normal CRMP2 band. Ratio of Upper shifted bands were significantly increased in *GLO1* (−/−) iPS cells (statistics: data are presented as the means ± SDs; *$P < 0.05$, and ***$P < 0.001$, as determined by one-way ANOVA followed by Tukey's multiple-comparisons test; n = 3).
Source data are available for this figure.

activity. Based on our basic molecular pathogenetic study of CRMP2 in the iPS cells under enhanced carbonyl stress, further in vivo analyses would deepen our understanding of the pathogenesis of schizophrenia in the developing brain.

To date, the expansion of irreversible protein aggregation in pathogenic contexts has been found in neurodegenerative diseases, including the aggregation of β-amyloid in Alzheimer's disease and the spreading of prion-aggregates in Creutzfeldt–Jakob disease. Here, we report that the aggregation and multimerization of AGE-CRMP2 forms one of the pathogenic pathway for a schizophrenia subset under carbonyl stress at an early stage (Fig 6). Recently, disrupted proteostasis resulting in protein misfolding or aggregation has been attributed to the etiopathogenesis of neuropsychiatric disorders, including schizophrenia (Bader et al, 2012; Korth, 2012; Hui et al, 2019). Although, the extent of aggregation was subtle in brain tissues from schizophrenia, when compared with the neurodegenerative disorders, mechanistic delineation of the process could provide crucial information on disease pathogenesis (Bradshaw & Korth, 2018). Interestingly, a related protein, CRMP1, was detected as aggregates, exclusively in the postmortem brain samples from schizophrenia (Bader et al, 2012). This evidence further underscores our observation of pathogenic multimeric aggregation of CRMP2, stemmed from enhanced carbonyl stress, possibly by altered proteostasis for the pathogenesis of schizophrenia. From this perspective, the expansion of protein aggregation, specifically in a developmental stage-dependent manner, might be the central cascade in both neurodegenerative diseases and psychiatric disorders. Future studies are warranted in postmortem samples from schizophrenia to show the subtle aggregation pathology involving CRMP2, particularly in the subset of patients with enhanced carbonyl stress (Ohnishi et al, 2019). Furthermore, the recent discovery for the role of mTOR signaling in the pathogenesis of neuropsychiatric disorders through altered proteostasis (Hui et al, 2019) and the regulation of the *DPYSL2* expression by mTOR (Liu et al, 2014; Pham et al, 2016) points to a possible additional role of dysregulated mTOR signaling in schizophrenia pathogenesis, through CRMP2-mediated aggregate formation. Because AGE modification induces the irreversible aggregation and polymerization of functional proteins, the development of therapeutics against carbonyl stress and diagnostic tools for the initial aggregation under carbonyl stress might be particularly important.

In summary, this report provides the first direct evidence showing CRMP2 as a target of enhanced carbonyl stress and revealed the mechanism of functional consequences of modified CRMP2 at the molecular and atomic levels at an early developmental stage. Further structural analyses in solution and physiological analyses in neurons and in vivo are strongly warranted.

# Materials and Methods

### Ethical approval

This study was approved by the Ethics Committees of RIKEN (approval No. wako3 2019-2) and all participating institutes and conducted according to the principles set out in the WDA Declaration of Helsinki and NIH Belmont Report. All participants gave informed and written consent to participate in the study.

### iPS cells

The experiments using iPS cells derived from two schizophrenia patients and one healthy control subject (Fig S1A) were approved by the Ethics Committees of RIKEN and were conducted in accordance to the principles expressed in the Declaration of Helsinki. The clinical histories of the patients have been described in our previous reports (Toyosima et al, 2011; Toyoshima et al, 2016). GLO1-deficient iPS cells were established in our preceding studies and cultured as described previously (Takahashi et al, 2007).

### Differentiation of iPS cells to the neuronal lineage

For the induction of neurospheres, iPS cells were dissociated into single cells using TrypLE Select (Life Technologies) and plated at a density of 10,000 cells/ml in an uncoated T75 flask containing the neural culture medium supplemented with SB431542, CHIR99021, dorsomorphin, hLIF, and bFGF (Fujimori et al, 2017). The cells were cultured in an atmosphere containing 4% $O_2$ and 5% $CO_2$ for 14 d.

### Neurosphere formation assay and neural differentiation

The dissociated cells were plated at a density of 10,000 cells in an uncoated 96-well plate, containing neural culture medium supplemented with hLIF and bFGF. The cells were then cultured in an atmosphere containing 4% $O_2$ and 5% $CO_2$ for 5 d, and the resulting neurosphere formation was analyzed. For neural differentiation, the neurospheres were dissociated into single cells and plated at a density of 10,000 cells per well on coverslips for 24-well plates coated with poly-L-ornithine (PO) (Sigma-Aldrich) and fibronectin (Sigma-Aldrich). For the induction of neuronal differentiation, the cells were further cultured for 3 d in the neural culture medium (without hLIF or bFGF) supplemented with 2% (vol/vol) B27 (Life Technologies), 10 ng/ml brain-derived neurotrophic factor (R&D Systems), 10 ng/ml glial-derived neurotrophic factor (R&D Systems), 200 $\mu M$ ascorbic acid (Sigma-Aldrich) and 1 mM dibutyryl-cAMP (Sigma-Aldrich).

### Western blot analysis

iPS cells and neurospheres were isolated, suspended in RIPA lysis buffer supplemented with a protease inhibitor cocktail (Sigma-Aldrich), triturated and centrifuged at 10,000$g$, and 4°C for 10 min. The supernatants were separated on 10% SDS–PAGE gels (10 $\mu g$ protein/lane) and blotted onto a polyvinylidene difluoride membrane. The membranes were then blocked with 5% skim milk, incubated with anti-AGE antibodies (CML: TransGenic, KH001, dilution 1/500; CEL: COSMO BIO, AGE-M02, dilution 1/1,000; MG-H1: CELL BIOLABS, STA-011, dilution 1/1,000; PEN: TransGenic, KH012, dilution 1/500; ARP: NOF CORPORATION, 5F6, dilution 1/500), or anti-GAPDH antibody (sc-20357, dilution 1/1,000; Santa Cruz Biotechnology) at 4°C overnight and then incubated with HRP-conjugated IgG (dilution 1/5,000; GE Healthcare) for 1 h at room temperature. The resulting signals were detected with an Immobilon Western Chemiluminescent

HRP Substrate (Merck MilliporeSigma), and the bands were analyzed using a FUSION Solo instrument (Vilber Lourmat). The intensities of the bands were quantified using Fusion software (Vilber Lourmat). The expression level of GLO1 was normalized to that of GAPDH.

## Immunoprecipitation

iPS cells were lysed in RIPA buffer containing a protease inhibitor cocktail (Nacalai Tesque) by sonication for 30 s. After centrifugation at 15,000$g$ for 10 min, the supernatant was incubated overnight at 4°C with anti-CRMP2 antibody (ab129082; Abcam) or normal rabbit IgG. Proteins were captured by protein G beads, eluted by boiling for 5 min in a sample buffer and analyzed by immunoblotting.

## Carbonyl stress assay

iPS cells were cultured in the presence of pyridoxamine (0, 5, 50, or 500 $\mu$M) for 4 d and lysed in RIPA buffer with sonication (30 s). The lysates (10 $\mu$g of protein) were analyzed by Western blotting using anti-AGE (TransGenic, KH001) and anti-GAPDH antibodies.

## Constructs, expression, and purification

The C532 construct (*Homo sapiens* CRMP2 residues 1–532 with a His tag) was bacterially expressed in the pET system with the pET-21b vector and *Escherichia coli* BL21 strain (Novagen). The harvested cells were lysed using a French Press (Ohtake Works) and centrifuged at 95,000$g$ for 30 min, and the supernatant was loaded on an Ni-NTA column (QIAGEN) with buffer A (30 mM Tris–HCl pH 7.5, 500 mM NaCl, 5% [vol/vol] glycerol, and 7 mM $\beta$-mercaptoethanol) and eluted with 500 mM imidazole in buffer A. The eluate was then further purified using a HiLoad 16/60 Superdex 200 SEC column (GE Healthcare) with buffer B (20 mM Tris–HCl pH 8.0, 150 mM NaCl, 1 mM DTT, and 5% [vol/vol] glycerol). To produce the N-terminal Myc-tagged CRMP2 construct for expression in iPS cells, full-length HsCRMP2 residues 1–572 were ligated into pCMV-Tag3A.

## Purification of AGE-CRMP2 in vivo and in vitro

For purification of endogenous CRMP2 from *GLO1*(−/−) iPS cells, cell lysates were centrifuged at 10,000$g$, and the supernatant was applied to a Mono Q column (GE Healthcare). The fractions containing CRMP2 protein were identified by dot blotting using anti-CRMP2 antibody. The obtained fraction containing CRMP2 protein was further separated using HiRes SEC columns (Tandem Superdex 200 Increased 10/300 GL columns; GE Healthcare) (Ogawa et al, 2017). The fractions containing CRMP2 were identified by dot blotting using anti-CRMP2 antibody. For the preparation of AGE-modified CRMP2 in vitro, recombinant C532 protein was incubated with 20 mM glyoxal in 20 mM PBS buffer at 20°C for 48 h. The AGE modification of both in vivo and in vitro protein samples was confirmed by Western blotting using anti-AGE antibody (TransGenic).

## Analyses of AGE sites by mass spectrometry

Both the endogenous CRMP2 purified from *GLO1*(−/−) iPS cells and the recombinant AGE-CRMP2 proteins were used for the mass

spectrometry analysis of AGE sites. For the derivatization of AGE sites, GRT (Yang et al, 2014) was added to each sample in 50 mM sodium acetate buffer (pH 4.5) at a final protein-to-GRT molar ratio of 1:10,000, and the samples were then incubated at room temperature for 16 h in the dark. The GRT-derivatized samples were reduced by sodium borohydride (NaBH$_4$) in 25 mM sodium phosphate buffer (pH 6.5) at room temperature for 1 h. After reduction in 4 M urea in 10 mM DTT/50 mM (NH$_4$)$_2$CO$_3$ solution at 50°C for 1 h and alkylation in 30 mM iodoacetamide/10 mM (NH$_4$)$_2$CO$_3$ for 30 min at room temperature in the dark, the samples were digested by Lys_C and Trypsin in Tris–HCl (pH 8.5) at 37°C for 16 h. To determine the AGE sites of CRMP2 in vitro and in vivo, each trypsinized peptide sample was analyzed by Q Exactive mass spectrometry with an Easy-nLC system (Thermo Fisher Scientific). Gradient nano-LC elution was performed with a 60-min gradient from 0.1% formic acid to 65% acetonitrile. The MS fragments were analyzed through an MS/MS ion search against the NCBI human database using Proteome Discoverer 1.4 software, including SequestHT (Thermo Fisher Scientific) and MASCOT (Matrix Science). The search was performed with the following parameters: a parent mass tolerance less than 5 ppm, a fragment mass tolerance less than 0.5 kD, a cutoff FDR of 5%, and a peptide rank of 1. Carbamidomethyl (C) was considered a static modification, and oxidation (M) and GRT-derivatization for Arg (72.057515 m/z), Lys (114.079313 m/z), Pro (131.105862 m/z), and Thr (113.095297 m/z) were considered dynamic modifications, with a set FRD of 5% (Fig 2).

## Crystallization, diffraction data collection, and processing

The crystallization of CRMP2 C532 protein was conducted at 287 K using the hanging-drop vapor diffusion method by mixing 1 $\mu$l of protein solution with 1 $\mu$l of reservoir solution. Crystals of un-modified and AGE-modified C532 proteins were obtained with 0.1 M calcium acetate hydrate, 0.1 M MES (pH 6.0), and 15% vol/vol PEG 400. The crystals were soaked in a cryoprotectant solution (20% [vol/vol] glycerol in reservoir solution) for 10 s, immediately flash-frozen, and preserved in liquid nitrogen for diffraction experiments. Diffraction data were collected at beamlines: BL5A and BL17A at the High Energy Accelerator Research Organization-Photon Factory (KEK-PF) and BL41XU at SPring-8. The datasets were then processed using HKL2000 (Otwinowski & Minor, 1997) and programs from the CCP4 package (Bailey, 1994). The statistical parameters for the data collection are summarized in Table 1.

## Structure determination, refinement, and analysis

Structure coordinates were determined by the molecular replacement method using MOLREP (Vagin & Teplyakov, 2000), and the structure of CRMP2 residues 13–516 (PDB code: 5MKV) (Zheng et al, 2018) was used as an initial search model. The models were further refined at 2.26- and 2.20-Å resolution, respectively, using Refmac5 (Murshudov et al, 2011) and COOT (Emsley & Cowtan, 2004) with manual model correction. After several cycles of interactive refinements, we successfully established final structure models with R$_{free}$/R$_{work}$ values of 0.171/0.193 and 0.183/0.204, and the final models were evaluated using the MOLPROBITY program (Davis et al, 2007). The data collection and refinement statistics are provided in Table 1. Structure-based multiple sequence alignment was performed

**Table 1. Diffraction data collection and refinement statistics.**

| | Unmodified C532 | AGE-C532 |
|---|---|---|
| Data collection statistics | | |
| X-ray source | PF-BL17A | PF-BL5A |
| Space group | $P2_1$ | $P2_1$ |
| Unit cell parameters | | |
| $a, b, c$ (Å) | 80.09, 158.43, 88.00 | 95.21, 147.33, 95.23 |
| $α, β, γ$ (°) | 90.00, 94.19, 90.00 | 90.00, 103.91, 90.00 |
| Wavelength (Å) | 1.0 | 1.0 |
| Resolution limits (Å)[a] | 50.00–2.26 (2.30–2.26) | 50.00–2.00 (2.06–2.00) |
| No. of unique reflections | 102,278 | 171,234 |
| Completeness (%) | 99.9 (99.9) | 100 (100) |
| Redundancy | 3.3 (3.3) | 3.8 (3.9) |
| $R_{merge}$ (%)[b] | 16.3 (57.9) | 10.4 (62.2) |
| $R_{p.i.m}$ (%) | 10.6 (37.5) | 6.2 (36.3) |
| Mean $I/σ$ (I) | 6.5 (2.5) | 9.1 (1.5) |
| CC1/2 | 0.96 (0.64) | 0.98 (0.71) |
| Refinement statistics | | |
| Resolution limits (Å) | 50.00–2.26 | 50.00–2.00 |
| $R_{work}$ (%)/$R_{free}$ (%)[c,d] | 17.1/19.3 | 18.3/20.4 |
| Rmsd for bonds (Å) | 0.014 | 0.010 |
| Rmsd for angles (°) | 1.507 | 1.360 |
| Wilson B-factor (Å$^2$) | 26.7 | 23.0 |
| No. of nonhydrogen protein atoms | 15,494 | 15,570 |
| No. of water oxygen atoms | 571 | 732 |
| Ramachandran plot (%) | | |
| Most favored regions | 96.80 | 96.17 |
| Additional allowed regions | 2.63 | 3.62 |
| Generously allowed regions | 0.57 | 0.21 |
| PDB entry | 6JV9 | 6JVB |

[a]Values in parentheses are for the highest resolution shell. About 5% of total reflections are used for highest resolution shell calculations.
[b]$R_{merge} = Σh\ Σl\ ||Ihl - ⟨Ih⟩|/Σh\ Σl\ ⟨Ih⟩$, where Ihl is the lth observation of reflection h and ⟨Ih⟩ is the weighted average intensity for all observations l of reflection h.
[c]$R_{work}$ factor $= Σh||Fobs(h)| - |Fcal(h)||/Σh|Fobs(h)|$, where Fobs(h) and Fcal(h) are the observed and calculated structure factors for reflection h, respectively.
[d]$R_{free}$ factor was calculated same as $R_{work}$ factor using the 5% the reflections selected randomly and omitted from refinement.

using the ESPript program (Robert & Gouet, 2014). The details of the interactions among residues of the structural interface were obtained using the Contact program in the CCP4 package (Bailey, 1994), and structural Fig were generated using the PyMOL program (DeLano Scientific LLC).

**HiRes-SEC assay**

HiRes-SEC assays were performed with a tandem connected column consisting of twin Superdex 200 Increase 10/300 columns (GE Healthcare) on an AKTA pure system (GE Healthcare) (Ogawa et al, 2017; Ogawa & Hirokawa, 2018). Purified unmodified or AGE-modified CRMP2 proteins (100 $μM$) were centrifuged, and the supernatants

(100 $μl$) were loaded onto the column containing assay buffer (20 mM Tris–HCl, pH 7.5, 150 mM NaCl, 1 mM DTT, and 5% [vol/vol] glycerol) for assessment. The eluted fractions, along with the pellets and supernatants of the total protein samples, were analyzed by SDS–PAGE. The data analysis was performed using Origin 8.0 (OriginLab Corp).

**DSC**

DSC was performed using a MicroCal PEAQ-DSC Automated system (Malvern Panalytical), which was temperature-calibrated with water and indium. The enthalpy was calibrated with indium. The protein samples (0.5 mg/ml) were applied into the cells of the calorimeter

and scanned at a heating rate of 200°C/h over the range of 20°C–120°C. Protein buffer was used as a reference. The thermograms acquired from the raw data after reference subtraction were fitted with baseline subtraction. Peak integration was performed using nonlinear least-square fitting, and the curve fit was further adjusted using the iterative curve fitting function included in the software provided with the instrument until a constant $\chi^2$ value was obtained.

### MT cosedimentation assay

Tubulin was purified from a porcine brain, and polymerized MTs were prepared using previously described standard methods (Ogawa & Hirokawa, 2015). Taxol-stabilized MTs (10 $\mu$M) were incubated with increasing concentrations of unmodified or AGE-modified CRMP2 proteins in BRB80K150 (80 mM PIPES, 150 mM KCl, and 1 mM $MgCl_2$) at 27°C for 15 min. Three independent assays were performed for each concentration, and the reaction samples were sedimented at a low speed (6,000$g$, 10 min) for measurement of the MT-bundling activity and a high speed (76,000$g$, 30 min) for assay of the total binding with MTs and then processed for SDS–PAGE analysis. The protein concentrations in the supernatant and pellet were quantified by densitometry of CBB-stained gels using ImageJ (NIH). The amounts of bound proteins and the proportions of bound proteins were plotted as functions of the amounts of unbound and total proteins, respectively. Because it was previously reported that one CRMP2-binding site is located on the tubulin-CRMP2 complex, the binding equation was fitted with the one-site model function using GraphPad Prism7 (GraphPad Software).

### MT-bundling assay and TIRF microscopy

The flow chambers for the MT-bundling assay were prepared with cover glass (MATSUNAMI) and spacers. The internal surface of the chamber was coated with NeutrAvidin (Thermo Fisher Scientific). The reaction mixture of polymerized MTs, 10% labelled with tetramethylrhodamine succinimidyl ester (TAMRA-X-SE, T-6105; Thermo Fisher Scientific) and 10% biotin succinimidyl ester (Biotium #90050; Biotin-SE) via standard MTs-labelling protocol (Gell et al, 2010), and CRMP2 (unmodified or AGE-modified CRMP2) (at a molar ratio of 1:4) was flowed into the chambers. The observations were performed using the ELYRA P.1 system (Carl Zeiss) with a 100× oil-immersion lens at 512 × 512 resolution in the TIRF mode. The MTs were randomly observed, and the length and intensity × area (1 × 2.5 $\mu$m) of the MTs were measured using ImageJ software (NIH). The statistical analyses were performed using GraphPad Prism7 (GraphPad Software).

### MT stability assay using iPS cells

GLO1 (+/+) and GLO1 (−/−) iPS cells were cultured as described above for 24 h after passage. Before fixation, nocodazole was added to the culture medium to destabilize unstable MTs. After 15 min of nocodazole treatment (1 $\mu$g/ml), the cells were fixed and permeabilized, and the MTs were stained with anti-$\alpha$ tubulin (DM1A) antibody (T6199, dilution 1/10,000; Sigma-Aldrich). The fluorescence intensity of the MTs was measured using ZEISS Efficient Navigation (Carl Zeiss).

## Data Availability

The coordinates for both the CRMP2 and AGE-CRMP2 structures have been deposited in the PDB database and are available under the accession codes 6JV9 (unmodified CRMP2) and 6JVB (AGE-CRMP2). The MS raw data and results files have been deposited in the ProteomeXchange Consortium (http://proteomecentral.proteomexchange.org) via the jPOST partner repository (http://jpostdb.org) (Okuda et al, 2017) under the dataset identifiers "jPOST:JPST00580" and "ProteomeXchange:PXD013522" (AGE-CRMP2 in vitro), and "jPOST:JPST00581" and "ProteomeXchange:PXD013524" (AGE-CRMP2 in iPS cells). All other data are available from the corresponding author on request.

## Supplementary Information

## Acknowledgements

This research was mainly supported by Japan Agency of Medical Research and Development (AMED) under grant numbers JP18dm0107083 to T Yoshikawa and JP18dm0908001 to N Hirokawa. The X-ray crystallography experiments were performed at beamlines BL-5A and BL-17A of KEK-PF under the approval of the Photon Factory Program Advisory Committee (proposal numbers 2018G143 and 2016G095) and at beamline BL41XU of SPring-8 with the approval of the Japan Synchrotron Radiation Research Institute (JASRI) (proposal numbers 2018A2552 and 2018B2552). The X-ray crystallography experiments were also supported by the Platform Project for Supporting Drug Discovery and Life Science Research (Basis for Supporting Innovative Drug Discovery and Life Science Research [BINDS]) from AMED under the grant number JP18am0101070, and the synchrotron radiation experiments were also performed at BL41XU of SPring-8 with the approval of JASRI (proposal numbers 2018A1003 and 2018B1011). The analysis of AGE-CRMP2 was partially supported by Japan Society for the Promotion of Science (JSPS) KAKENHI grant number 18K07616 to M Toyoshima. We thank Drs T Senda, N Matsugaki, Y Yamada, and M Senda and all the staff at the Photon Factory as well as Drs K Hasegawa, N Mizuno, and H Murakami and all the staff at SPring-8 for the help with the X-ray crystal data collection. We also thank all the members of T Yoshikawa's laboratory and N Hirokawa's laboratory for the valuable discussions and help.

### Author Contributions

M Toyoshima: conceptualization, resources, data curation, validation, investigation, and writing—original draft, review, and editing.
X Jiang: conceptualization, data curation, validation, and writing—original draft, review, and editing.
T Ogawa: conceptualization, data curation, supervision, validation, and writing—original draft, review, and editing.
T Ohnishi: resources, investigation, and writing—review and editing.
S Yoshihara: investigation and writing—review and editing.
S Balan: investigation and writing—review and editing.
T Yoshikawa: conceptualization, data curation, supervision, funding acquisition, validation, investigation, project administration, and writing—original draft, review, and editing.

N Hirokawa: conceptualization, data curation, supervision, funding acquisition, investigation, project administration, and writing—original draft, review, and editing.

## Conflict of Interest Statement

The authors declare that they have no conflict of interest.

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
