## [Reviewer comments · Life Science Alliance]

Life Science Alliance

Enhanced Carbonyl Stress Induces Irreversible Multimerization of CRMP2 in Schizophrenia Pathogenesis

Manabu Toyoshima, Xuguang Jiang, Tadayuki Ogawa, Tetsuo OHNISHI, Shogo Yoshihara, Shabeesh Balan, Takeo Yoshikawa, and Nobutaka Hirokawa

DOI: <https://doi.org/10.26508/lsa.201900478>

Corresponding author(s): Nobutaka Hirokawa, Univ Tokyo Grad Sch Med and Takeo Yoshikawa, RIKEN Center for Brain Science

Review Timeline:

Submission Date:	2019-07-01
Editorial Decision:	2019-07-18
Revision Received:	2019-08-15
Editorial Decision:	2019-09-11
Revision Received:	2019-09-17
Accepted:	2019-09-19

Scientific Editor: Andrea Leibfried

Transaction Report:

July 18, 2019

Re: Life Science Alliance manuscript #LSA-2019-00478-T

Prof. Nobutaka Hirokawa
Univ Tokyo Grad Sch Med
Dept Mol Cell Biology
7-3-1 Hongo
7-3-1 Hongo
Bunkyo-ku, Tokyo J-Tokyo 113-0033
Japan

Dear Dr. Hirokawa,

Thank you for submitting your manuscript entitled "Enhanced Carbonyl Stress Induces Irreversible Multimerization of CRMP2 in Schizophrenia Pathogenesis" to Life Science Alliance. The manuscript was assessed by expert reviewers, whose comments are appended to this letter.

As you will see, the reviewers appreciate your findings and provide constructive input on how to strengthen your study to allow publication here. We would thus like to invite you to submit a revised version to us, addressing the individual concerns raised by the reviewers. This seems rather straightforward, but please do get in touch with us in case you would like to discuss individual revision points further.

Thank you for this interesting contribution to Life Science Alliance. We are looking forward to receiving your revised manuscript.

Sincerely,

Andrea Leibfried, PhD
Executive Editor
Life Science Alliance
Meyershofstr. 1
69117 Heidelberg, Germany
t +49 6221 8891 502
e a.leibfried@life-science-alliance.org
www.life-science-alliance.org

B. MANUSCRIPT ORGANIZATION AND FORMATTING:

Reviewer #1 (Comments to the Authors (Required)):

The authors investigate the targets of carbonyl stress by a dysfunctional GLO1 enzyme in differentiated IPS cells and discover CRMP2 as a major target. They show the AGE modified sites by structural studies and present a model of irreversible multimerization as a cause for functional

effects on microtubules. They conclude that this could be a novel mechanism for schizophrenia.

The paper is new, exciting, and highly relevant. It covers a range of meticulously performed methods and presents a novel timely model of disease-relevant posttranslational protein modification in a field (molecular psychiatry) that, so far, has lacked rigorous biochemical studies of the kind presented here. The data support the conclusions.

I have only minor issues that the authors should change in the manuscript:

- in the introduction, a broader background on oxidative stress and schizophrenia should be given, e.g. quote Steullet et al. 2016 Schiz Res and related research

- methods: a weakness is that no congenic iPSC controls were used but I believe the presented effects are sufficiently strong; nevertheless this should be critically discussed. Also: how many independent differentiations were done?

- methods: in the listing of antibodies, ordering/ batch numbers from providers used should be designated

- in the discussion and Figure 6: the authors should not claim that they have discovered a new mechanism for schizophrenia which assumes ALL cases schizophrenia. It is generally believed and there is abundant evidence that schizophrenia is a heterogeneous condition where different biological mechanisms converge into one clinical diagnosis. I propose therefore that the authors use the word "schizophrenia subset" instead.

- in Figure 1C, J, L molecular size markers should be indicated

Reviewer #2 (Comments to the Authors (Required)):

"Enhanced Carbonyl Stress Induces Irreversible Multimerization of CRMP2 in Schizophrenia Pathogenesis"

By M. Toyoshima et al.

Reactive stress causes carbonylation (AGE modification) of proteins, which has been implicated in the neurodevelopmental disorder schizophrenia. Using iPSC cells, the authors identify the axon growth/guidance factor CRMP2 as major target for carbonylation and use protein biochemistry and structural biology to elucidate how the AGE modifications inactivate CRMP2.

This manuscript is an interesting addition to our understanding of neurodevelopmental diseases. Still, I think that some points should be considered before publishing the manuscript:

1. First of all, the recombinant AGE-CRMP2 needs better description. According to Materials & Methods, purified CRMP2 was reacted with glyoxal, i.e., the protein was not modified in the GLO1(-/-) cell system of the first Figures. It is therefore crucial to understand if (i) the same residues are modified, and (ii) if the modification stoichiometry of the recombinant protein matches the in vivo-modified protein.

2. Related: the crystal structure of AGE-CRMP2 at the bottom of Figure 4B may show all detected

AGE modifications, but a rigorous - e.g., tabular - description needs to be added. Are all modifications defined by sufficient electron density to deduce the local structure?

3. Finally, I assume that in vivo, the AGE modification load is less than after in-vitro modification with glyoxal. If this is the case, the authors should discuss how much the mechanism described in the second half of the paper applies to the naturally modified CRMP2.

4. Why do the GLO1(+fs) patient's iPS cells show a stronger phenotype than the artificial GLO1(-/-) cells? The authors should discuss if the GLO1(fs) allele may have a toxic effect, or if there is a second-hit mutation in the patient's iPS cells.

5. The text describes in several instances a protein preparation as "roughly purified". May I suggest using the term "enriched" instead?

6. At the top of page 9, the anion exchange is described as "AKTA system, GE healthcare". For the main text, it would be more interesting to name the column material (MonoQ) instead of the chromatography apparatus.

7. At the middle of page 9, it is concluded that the data "revealed that the impaired cellular and developmental phenotypes [...] stemmed from the carbonylation of CRMP2". This conclusion is too strong and should be modified: while this reviewer agrees that the major modified protein is CRMP2, the phenotypes could be caused, partially or entirely, by minor modified proteins that are not identified here.

8. At the bottom of page 10, Fig. 2D is cited for the high-accuracy determination of the AGE site on K390. Most likely, the authors mean Fig. 2B. Fig. 2B, however, needs to be better described to be understandable. Also, it will be difficult to reproduce the panels at the given size in sufficient quality - the authors should consider giving the panels more space in the Figure.

9. Since the crystallography part is described only from Fig. 4 on, I assume that the structure shown in Fig. 2C/D is one of the previously solved CRMP2 structures, isn't it? In this case, it should be properly cited. Similarly, the CRMP2 structure 5MKV that was used to solve the structures here should also be acknowledged with a proper citation.

10. How were the MTs labelled for Fig. 3A/B?

11. Page 13 cites Fig. 3C for an MT turbidity assay. Actually, Fig. 3C does not show turbidity data. The authors should correct this sentence.

12. The very top of page 14 ("By comparing over all tetrameric complexes, apparent structural differences...") seems to imply that the crystallographical unit cell contains a tetramer. Is this true? Perhaps the crystallography findings could be described a bit better.

13. In Fig. 5C, what is the identity of the lower-shifted bands? In addition, it appears as if not only the higher-shifted bands vary in intensity, but also the main CRMP2 band. Therefore, to have a conclusive panel, the authors should scan the image, calculate the ratio of the shifted bands over the main band, and do proper statistics.

Response to the comments of reviewer #1:

The authors investigate the targets of carbonyl stress by a dysfunctional GLO1 enzyme in differentiated IPS cells and discover CRMP2 as a major target. They show the AGE modified sites by structural studies and present a model of irreversible multimerization as a cause for functional effects on microtubules. They conclude that this could be a novel mechanism for schizophrenia. The paper is new, exciting, and highly relevant. It covers a range of meticulously performed methods and presents a novel timely model of disease-relevant posttranslational protein modification in a field (molecular psychiatry) that, so far, has lacked rigorous biochemical studies of the kind presented here. The data support the conclusions.

Response: Thank you very much for your brilliant digestion and favorable consideration of our manuscript.

I have only minor issues that the authors should change in the manuscript:

1. - in the introduction, a broader background on oxidative stress and schizophrenia should be given, e.g. quote Steullet et al. 2016 SchizRes and related research

Answer: Thank you for your pertinent suggestion. We have expanded the description on this aspect with quoting the relevant papers including Steullet et al. 2016 SchizRes to introduce a broader background on oxidative stress and schizophrenia in the introduction section of our

revised manuscript. As per the suggestions of the reviewer we have incorporated following sentences in the introduction (in P. 4; para. 2):

“Oxidative stress, which has been suggested to be crucial for the pathogenesis of schizophrenia, mainly causes oxidative damage to the lipids, proteins, and DNA (Bitanirwe & Woo, 2011). It also forms one of the central hub systems, which when perturbed affects the integrity of parvalbumin interneurons and oligodendrocytes, a characteristic feature observed in schizophrenia (Steullet et al, 2016).”

2. - methods: a weakness is that no congenic iPSC controls were used but I believe the presented effects are sufficiently strong; nevertheless this should be critically discussed. Also: how many independent differentiations were done?

Answer: In this manuscript, we initially showed that the schizophrenia patient-derived hiPSCs with *GLO1* frameshift mutation formed lesser number of neurospheres, which was rescued by the treatment of reactive carbonyl compound scavenger pyridoxamine. We further generated *GLO1*-deficient hiPSC lines from healthy controls and confirmed that the reduction in number of neurospheres were persistent in *GLO1*-deficient hiPSC lines, when compared to the isogenic controls. These deficits in cellular phenotypes could also be rescued by the pyridoxamine treatment. These results further underscored that the defective cellular phenotypes observed was due to the elevated the level of enhanced carbonyl stress resulting from *GLO1* loss of function. Since schizophrenia patient-derived hiPSCs with *GLO1* frameshift mutation showed more pronounced deficits in cellular phenotypes, we reasoned that this might be due to the presence of additional schizophrenia associated genetic risk factors in the subject, which might further enhance the observed effect (Balan et al 2014, Toyoshima et al 2016).

3. - methods: in the listing of antibodies, ordering/ batch numbers from providers used should be designated

Answer: Thanks for pointing out this issue. We have added the ordering/batch numbers from providers of the antibodies in the revised version of our manuscript.

*4. - in the discussion and Figure 6: the authors should not claim that they have discovered a new mechanism for schizophrenia which assumes ALL cases schizophrenia. It is generally believed and there is abundant evidence that schizophrenia is a heterogenous condition where different biological mechanisms converge into one clinical diagnosis. I propose therefore that the authors use the word "schizophrenia subset" instead.*

**Answer:** Thanks for your constructive advice. We have modified relevant portions in the discussion and the figure legend of figure 6 according to the reviewer's recommendation (in P. 37).

5. *- in Figure 1C, J, L molecular size markers should be indicated*

**Answer:** Many thanks for your kind reminding. The Figure 1C, J, L has been revised by indicating the molecular size marker.

Response to the comments of reviewer #2:

"Enhanced Carbonyl Stress Induces Irreversible Multimerization of CRMP2 in Schizophrenia Pathogenesis" By M. Toyoshima et al.

Reactive stress causes carbonylation (AGE modification) of proteins, which has been implicated in the neurodevelopmental disorder schizophrenia. Using iPS cells, the authors identify the axon growth/guidance factor CRMP2 as major target for carbonylation and use protein biochemistry and structural biology to elucidate how the AGE modifications inactivate CRMP2. This manuscript is an interesting addition to our understanding of neurodevelopmental diseases. Still, I think that some points should be considered before publishing the manuscript:

Response: Thank you very much for your precise evaluation and favorable consideration of our manuscript.

1. First of all, the recombinant AGE-CRMP2 needs better description. According to Materials & Methods, purified CRMP2 was reacted with glyoxal, i.e., the protein was not modified in the GLO1(-/-) cell system of the first Figures. It is therefore crucial to understand if (i) the same residues are modified, and (ii) if the modification stoichiometry of the recombinant protein matches the in vivo-modified protein.

Answer: Thank you for your valuable comments and suggestions. Due to the lack of GLO1 protein that functions in the metabolism of methylglyoxal, the methylglyoxal level as well as the carbonyl stress highly increased in GLO1(-/-) cell system. Therefore the proteins can be also AGE-modified in the GLO1(-/-) cell system, with the evidences shown in Figure 1I, 1L and Figure S1B. And therefore, we sought to use *in vitro*-glyoxal inducement to mimic the *in vivo* AGE modification of protein. Our further results of LC/MS assessment of the AGE sites of CRMP2 using both *in vivo* and *in vitro* samples showed that the detected *in vivo* AGE sites matches well with the detected *in vitro*-glyoxal induced AGE sites. A total of 50 and 62 AGE sites (GRT-derivatizations of lysine, arginine, threonine and proline) were detected for *in vivo* and *in vitro* samples, respectively, with 29 sites overlapped (7 arginines, 11 lysines 5 prolines, and 6 threonines). And extra 12 and 14 sites situate very closely to the overlapping sites (± 5

residues) (Figure 2A, S2A-B). We have revised the relevant text to provide a clearer description (in P. 12; para. 1):

“...which suggest that the *in vitro* glyoxal-treated CRMP2 highly resembles the *in vivo* AGE-modified CRMP2 in terms of the distribution of AGE sites and thereby is appropriate to be utilized for further *in vitro* analysis.”

As for your next concern about the modification stoichiometry, since it is currently difficult to achieve the site-directed quantitation for over 60 sites by quantitative LC/MS analysis of a protein, we instead estimated the *in vivo* and *in vitro* stoichiometry of AGE-CRMP2 from WB and SDS-PAGE images shown in Figure 5C and Figure S1E, respectively. And also, we do consider that this issue need to be further discussed in our manuscript. We have added more discussion concerning this issue in the revised version of our manuscript at discussion section (in P. 18; para. 2):

“In this paper, we analyzed the AGE-sites of CRMP2 both in iPS cells and *in vitro*. By using LC/MS analysis, we qualitatively determined over 60 AGE-sites. The quantitation of stoichiometry distribution of each AGE-site in iPS cells and post-mortem brain should be further investigated in future. Based on the comparison of our data in iPS cells (Fig 5C and D) and *in vitro* (Fig S1E), we speculate that there is a similar stoichiometry of AGE-modification between our iPS cells-derived and *in vitro* glyoxal-induced CRMP2 proteins.”

2. Related: the crystal structure of AGE-CRMP2 at the bottom of Figure 4B may show all detected AGE modifications, but a rigorous - e.g., tabular - description needs to be added. Are all modifications defined by sufficient electron density to deduce the local structure?

Answer: Thanks for your valuable comments and suggestion. We used high sensitive LC/MS technique to successfully detect the AGE-modified residues of CRMP2. As we commented at the previous part (1.), there is a stoichiometry variety between *in vivo* (Fig 5D) and *in vitro* (Fig S1E) AGE-modified sites, both of which could not achieve a high modification ratio for each AGE-site. On the other hand, the diffraction data represent the averaged electric density map, therefore, only several AGE-sites show strong electron density enough to be identified

as the AGE-modified residue. And all these sites shown in Figure 4B that exhibit a significant change on density map corresponds with a high intensity in the LC/MS assessment both *in vivo* and *in vitro*.

We have revised the relevant part in the text to provide more discussion concerning this issue (in P. 18; para. 2).

3. Finally, I assume that *in vivo*, the AGE modification load is less than after *in-vitro* modification with glyoxal. If this is the case, the authors should discuss how much the mechanism described in the second half of the paper applies to the naturally modified CRMP2.

Answer: Thanks for your valuable comments and constructive suggestion. We understand your assumption that the AGE modification load *in vivo* is probably less than after *in-vitro* modification with glyoxal. We have added the quantitative analyses of band-shift ratio of AGE-CRMP2, and compared the differences in stoichiometry between *in vitro* (Fig S1E) and *in vivo* conditions (Fig 5D). More discussion concerning this issue has been documented at the discussion section of our revised manuscript (in P. 18; para. 2).

4. Why do the *GLO1(+fs)* patient's iPS cells show a stronger phenotype than the artificial *GLO1(-/-)* cells? The authors should discuss if the *GLO1(fs)* allele may have a toxic effect, or if there is a second-hit mutation in the patient's iPS cells.

Answer: We showed that defective cellular phenotypes observed in neural progenitors cell and neurons were due to the elevated the level of enhanced carbonyl stress resulting from *GLO1* loss of function. However, schizophrenia patient-derived hiPSCs with *GLO1* frameshift mutation showed more pronounced deficits in cellular phenotypes. We reasoned that this might be due to the presence of additional schizophrenia associated genetic risk factors in the subject, which we reported previously (Toyoshima et al., 2016). We have incorporated this information in P. 9 (para. 1).

5. The text describes in several instances a protein preparation as "roughly purified". May I

suggest using the term "enriched" instead?

Answer: Thanks for your helpful suggestion and sorry for our inappropriate description. We have revised our manuscript by replacing the “roughly purified” with the term “enriched” (in Pp. 10-11).

6. At the top of page 9, the anion exchange is described as "AKTA system, GE healthcare". For the main text, it would be more interesting to name the column material (MonoQ) instead of the chromatography apparatus.

Answer: Thanks for your pertinent suggestion and we totally agree with your opinion. We have revised the description to name the column we used (MonoQ) instead of the chromatography apparatus in the main text (in P. 10; para. 2).

7. At the middle of page 9, it is concluded that the data "revealed that the impaired cellular and developmental phenotypes [...] stemmed from the carbonylation of CRMP2". This conclusion is too strong and should be modified: while this reviewer agrees that the major modified protein is CRMP2, the phenotypes could be caused, partially or entirely, by minor modified proteins that are not identified here.

Answer: Thanks for your helpful suggestion. We have modified this concluding sentence to clearly indicate that the impaired cellular and developmental phenotypes are achieved not only by the effect of carbonylation of CRMP2, but also that of carbonylation of other minor unidentified proteins. We have modified the above statement taking into consideration of the reviewer’s suggestion as follows (in P. 10; para. 2):

“We revealed that the impaired cellular and developmental phenotypes manifested in the neural cells derived from *GLO1*-deficient iPS cells were stemmed mainly from the carbonylation (CML) of CRMP2. However, the role of other carbonylated proteins contributing to these observed phenotypes in a synergistic manner can also not be excluded.”

8. At the bottom of page 10, Fig. 2D is cited for the high-accuracy determination of the AGE site on K390. Most likely, the authors mean Fig. 2B. Fig. 2B, however, needs to be better

described to be understandable. Also, it will be difficult to reproduce the panels at the given size in sufficient quality - the authors should consider giving the panels more space in the Figure.

Answer: Thanks for your careful scrutiny and good suggestions. We have corrected the miscitation of the figure and we have also improved the description and presentation of the Fig. 2B to make it more understandable. More space and higher resolution images have been provided for the Fig. 2B.

And the raw data and results files can be assessed directly in the ProteomeXchange Consortium (<http://proteomecentral.proteomexchange.org>) via the jPOST partner repository (<http://jpostdb.org>) under the dataset identifiers JSPT000580/PXD013522 (AGE-CRMP *in vitro*) and JSPT000581/PXD013524 (AGE-CRMP in iPS cells).

9. Since the crystallography part is described only from Fig. 4 on, I assume that the structure shown in Fig. 2C/D is one of the previously solved CRMP2 structures, isn't it? In this case, it should be properly cited. Similarly, the CRMP2 structure 5MKV that was used to solve the structures here should also be acknowledged with a proper citation.

Answer: Thanks for your careful scrutiny and helpful suggestions. The structure shown in Fig. 2C is the structure model of 5MKV because it achieves the best visualization of C-terminus of CRMP2 among current structure models of CRMP2 so that we used it for *in silico* representation of the distribution of AGE-sites on CRMP2. And for a better comparison between unmodified and AGE-modified CRMP2, we used the structure models we solved in this work (6JV9 & 6JVB) to perform the simulation shown in Fig 2D. We have revised the text of the figure legends to indicate the resource of the structure models we used for presentation clearly (in P. 33). And also, we have revised the relevant text to cite the paper that reported 5MKV (in P. 26; para. 3).

10. How were the MTs labelled for Fig. 3A/B?

Answer: Thank you for your question. The MTs that we used here were prepared with a mixture of rhodamine-labelled and biotin-labelled tubulins.

We have revised the relevant part in the Materials & Methods section in our revised paper (in P. 28; para. 2):

“The reaction mixture of polymerized MTs, 10% (labelled with tetramethylrhodamine succinimidyl ester (TAMRA-X-SE, Thermofisher, T-6105) rhodamine and 10% biotin succinimidyl ester (Biotin-SE, Biotium #90050) via standard MTs-labeling protocol (Gell et al, 2010)”

11. Page 13 cites Fig. 3C for an MT turbidity assay. Actually, Fig. 3C does not show turbidity data. The authors should correct this sentence.

Answer: Thanks for your careful scrutiny. The “Fig. 3C” here should be Fig. S3C and we have corrected this sentence (in P. 14).

12. The very top of page 14 ("By comparing over all tetrameric complexes, apparent structural differences...") seems to imply that the crystallographical unit cell contains a tetramer. Is this true? Perhaps the crystallography findings could be described a bit better.

Answer: Thanks for your question and pertinent suggestion. We feel sorry that there is actually a typo here: the “over all” should be “overall”. What we would like to indicate here is that we compared structures of unmodified and AGE-modified CRMP2 in tetrameric conformation (functional state of CRMP2 with MTs-bundling activity) and found significant structure differences. In addition, our solved crystal structures of CRMP2 both exhibit a tetramer in one crystallographical asymmetric unit.

We have revised this sentence for a clearer description in the revised manuscript (in P. 15; para. 2):

“By comparing the overall tetrameric complex structures, apparent structural differences were observed in the N-terminal region, especially for the $\beta 2$ and $\beta 3$ sheets that include the AGE-target K23 in T-sites.”

13. In Fig. 5C, what is the identity of the lower-shifted bands? In addition, it appears as if not

only the higher-shifted bands vary in intensity, but also the main CRMP2 band. Therefore, to have a conclusive panel, the authors should scan the image, calculate the ratio of the shifted bands over the main band, and do proper statistics.

Answer: Thank you for your valuable question and suggestion. We consider that the lower-shifted bands may probably be the degradation fragment of CRMP2 protein under carbonyl stress. And based on your advice, we have scanned the image, calculated and performed statistical analysis. More discussion has been also documented in the discussion section of our revised manuscript (in P. 18; para. 2):

“In this paper, we analysed the AGE-sites of CRMP2 both in iPS cells and *in vitro*. By using LC/MS analysis, we qualitatively determined over 60 AGE-sites. The quantitation of stoichiometry distribution of each AGE-site in iPS cells and post-mortem brain should be further investigated in future. Based on the comparison of our data in iPS cells (Fig 5D) and *in vitro* (Fig S1E), we speculate that there is a similar stoichiometry of AGE-modification between our iPS cells-derived and *in vitro* glyoxal-induced CRMP2 proteins.”

Fig. 5C, D

Fig. S1E

September 11, 2019

RE: Life Science Alliance Manuscript #LSA-2019-00478-TR

Prof. Nobutaka Hirokawa
Univ Tokyo Grad Sch Med
Dept Mol Cell Biology
7-3-1 Hongo
Bunkyo-ku, Tokyo J-Tokyo 113-0033
Japan

Dear Dr. Hirokawa,

Thank you for submitting your revised manuscript entitled "Enhanced Carbonyl Stress Induces Irreversible Multimerization of CRMP2 in Schizophrenia Pathogenesis". As you will see, reviewer #2 appreciates the introduced changes and we would thus be happy to publish your paper in Life Science Alliance pending final revisions necessary to meet our formatting guidelines:

- please include the supplementary info in the main manuscript file
- please link your ORCID iD to your profile in our submission system
- please add a callout to Fig 1D
- please refer to current fig S4 as a "table" and provide it in either word or excel format
- please provide the source data for the GAPDH blots in FigS1B

A. FINAL FILES:

- An editable version of the final text (.DOC or .DOCX) is needed for copyediting (no PDFs).
- High-resolution figure, supplementary figure and video files uploaded as individual files: See our detailed guidelines for preparing your production-ready images, <http://www.life-science-alliance.org/authors>
- Summary blurb (enter in submission system): A short text summarizing in a single sentence the

study (max. 200 characters including spaces). This text is used in conjunction with the titles of papers, hence should be informative and complementary to the title. It should describe the context and significance of the findings for a general readership; it should be written in the present tense and refer to the work in the third person. Author names should not be mentioned.

B. MANUSCRIPT ORGANIZATION AND FORMATTING:

Sincerely,

Reviewer #2 (Comments to the Authors (Required)):

In the revised version, the authors have modified the manuscript to better explain/discuss the results and added a quantification of Western blots. My concerns are sufficiently addressed in the new version of the manuscript, and I see it fit for publication.

September 19, 2019

RE: Life Science Alliance Manuscript #LSA-2019-00478-TRR

Prof. Nobutaka Hirokawa
Univ Tokyo Grad Sch Med
Dept Mol Cell Biology
7-3-1 Hongo
Bunkyo-ku, Tokyo J-Tokyo 113-0033
Japan

Dear Dr. Hirokawa,

Thank you for submitting your Research Article entitled "Enhanced Carbonyl Stress Induces Irreversible Multimerization of CRMP2 in Schizophrenia Pathogenesis". It is a pleasure to let you know that your manuscript is now accepted for publication in Life Science Alliance. Congratulations on this interesting work.

DISTRIBUTION OF MATERIALS:

Again, congratulations on a very nice paper. I hope you found the review process to be constructive and are pleased with how the manuscript was handled editorially. We look forward to future exciting submissions from your lab.

Sincerely,

Andrea Leibfried, PhD
Executive Editor
Life Science Alliance
Meyerohofstr. 1
69117 Heidelberg, Germany
t +49 6221 8891 502
e a.leibfried@life-science-alliance.org
www.life-science-alliance.org